# MAGE: Leveraging LLMs for Automated Mapper Generation in Parallel Programming

## Abstract

Efficiently mapping tasks to processors and data to memories is a cornerstone of parallel programming to achieve high performance. Traditionally, this critical task has been handled by expert-crafted mapper programs, tailored for specific machine architectures and problem domains. However, creating customized mappers for each unique application is labor-intensive and time-consuming. Large language models (LLMs) have recently demonstrated remarkable capabilities in understanding and generating code, as well as in self-improvement for optimizing specific performance metrics. Inspired by these advancements, we introduce the task of **ma**pper **ge**neration (MAGE), which frames generating high-performance mappers as a discrete optimization problem aimed at maximizing compute throughput. To solve this optimization problem, we leverage agentic LLMs in the mapper generation process. At the core of our approach lies a novel domain-specific language (DSL), which provides a high-level interface for LLMs to generate the mapper code without getting entangled with complicated, low-level system programming. Moreover, our DSL defines a structured and constrained search space for RL to explore, guiding LLMs to discover the optimal mapping policy. The evaluation shows that our LLM-generated mappers can surpass expert-written mappers in performance, achieving up to 34% speedup across 9 benchmarks. Notably, our approach improves the throughput of parallel matrix multiplication algorithms by up to 31%, reducing development time from several days to just a few minutes.

## 1 Introduction

Task-based programming (Slaughter et al., 2015; Bauer et al., 2012; Augonnet et al., 2009; Chamberlain et al., 2007; Kaiser et al., 2014; Heller et al., 2017; Chandra et al., 2001; Duran et al., 2011; Moritz et al., 2018; Barham et al., 2022) has recently emerged as a prominent paradigm in parallel programming. The core idea is to decompose computations into self-contained functions called *tasks* that do not communicate with other tasks except through their *collection* (typically tensor or multi-dimensional array) arguments or data.

One performance-critical aspect of executing task-based applications is *mapping*: assigning tasks to processors, data to physical memories, and managing other low-level physical resources. We refer to a concrete mapping policy as a *mapper*. Different from directly modifying application-level (e.g., CUDA-level) code, mappers operate at a higher level (e.g., task and processor level) and do not change the correctness of an application's output; they only affect its performance. The difference between a good mapper and a bad mapper is easily an order of magnitude or even more in performance. Currently, writing mappers is a manual, labor-intensive, and time-consuming process that requires deep knowledge of the application, the machine, its resource limits, and the C++ mapping APIs. Even for experienced performance engineers, developing an effective mapper can take several days due to the complexity and iterative refinement required.

In this paper, we focus on using LLMs to automatically generate mappers. There are two main challenges in using LLMs for mapper code generation. First, when given a specific mapping policy described in natural language, LLMs struggle to generate the corresponding mapper code, which requires generating a few hundred lines of low-level C++ code. This code generation task contrasts significantly with method- or function-level code generation, which is more isolated and where

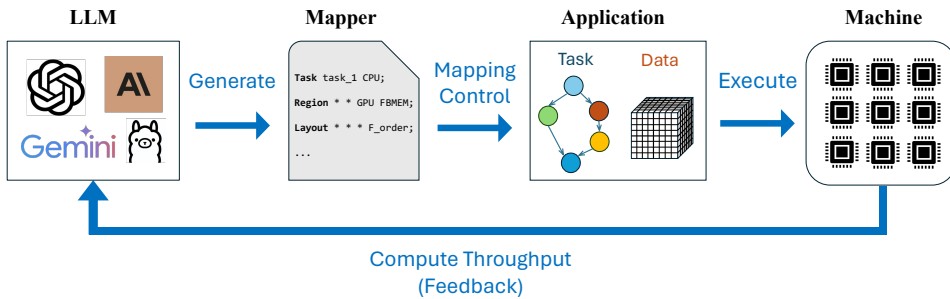

Figure 1: Agent-based search for mapper code generation, where the mapper is written in a Domain-Specific Language (DSL) and refined through reinforcement learning (RL) with interactive feedback to maximize application performance.

LLMs have demonstrated strong performance (Chen et al., 2021; Austin et al., 2021). In our case, generating C++ mappers requires a deep understanding of low-level systems and broader contextual dependencies in large-scale system code Jimenez et al., which is beyond the capabilities of current LLMs (Du et al., 2023; Wang et al., 2024).

To address the first challenge, we design a self-contained Domain-Specific Language (DSL) for mapper generation. Unlike the C++ APIs, the DSL provides a *high-level interface*, allowing LLMs to generate code without handling the low-level intricacies and complexities of C++ APIs.

The second challenge is to identify the optimal mapping strategy that maximizes the performance of application programs. Our key insight is to reformulate mapper generation into a search problem, a category of problems that is well-suited to agentic LLMs that are inspired by classical reinforcement learning (RL) techniques. Notably, RL has achieved tremendous success across a wide range of applications, such as AlphaTensor (for finding better matrix multiplication) (Fawzi et al., 2022) and AlphaGeometry (for generating math proofs for geometry problems) (Trinh et al., 2024). For the mapper generation problem, our DSL constructs a more *structured and constrained search space* for LLMs to explore.

For the search process, we combine the LLM's code generation capabilities with an agent-based solution built on the Trace framework (Cheng et al., 2024). In this approach, LLMs act as optimizers, generating mappers in the DSL and receiving iterative feedback through an interactive feedback mechanism as shown in Figure 1. This feedback loop allows LLMs to refine their mapping strategies efficiently, reducing the time required to develop high-performance mappers to just a few minutes. Given the context-specific nature of mappers, this agent-based RL approach is particularly advantageous. Our experiments show that mappers generated by LLMs not only match but can surpass expert-written mappers, achieving up to 34% speedup across 9 benchmarks. Notably, the search process enhances the performance of parallel matrix multiplication algorithms by up to 31%. We also demonstrate, through ablation studies, that the critical role of high-quality feedback in guiding LLM optimizers to discover more efficient mapping strategies.

1. **Development of a Domain-Specific Language (DSL) to Simplify Mapper Generation:** We design a DSL to address the complexities of direct code generation in low-level systems. This DSL provides a higher-level abstraction, enabling LLMs to generate mapping code more effectively by encapsulating the complexities of low-level C++ APIs.

2. **Formulation of Mapper Generation as a Discrete Optimization Problem:** We formulate the task of **ma**pper **ge**neration (MAGE) as a discrete optimization problem with the objective of maximizing application performance. Our DSL defines a more structured and constrained search space for LLM to explore.

3. **Experimental Validation and Performance Comparison:** Our DSL significantly improves the success rate of mapper code generation for a given mapping strategy from 0% in C++ to 80% in our DSL. Leveraging high-quality feedback, our agent-based solution discovers mappers that achieve up to 34% speedup compared with expert-designed mappers, enhancing the performance of parallel matrix multiplication algorithms by up to 31%.

## 2 RELATED WORK

**Mapping in Parallel Programming** Many parallel programming systems allow users to make their own mapping decisions, such as Legion (Bauer et al., 2012), StarPU (Augonnet et al., 2009; 2010), Chapel (Chamberlain et al., 2007), HPX (Kaiser et al., 2014; Heller et al., 2017), Sequoia (Fatahalian et al., 2006), Ray (Moritz et al., 2018), TaskFlow (Huang et al., 2021), and Pathways (Barham et al., 2022). Several techniques have been proposed to automate mapping, including machine learning models (O'Boyle et al., 2013; Wang & O'Boyle, 2009), static analysis (Poesia et al., 2017; Ren et al., 2008), traditional reinforcement learning techniques (Mirhoseini et al., 2017), and autotuning (SFX Teixeira et al., 2023). Unlike previous work, we use an agent-based reinforcement learning approach with LLMs, exploring a larger search space for mappers than traditional methods.

**LLM Code Generation** With the rise of LLMs, various models (Chen et al., 2021; Nijkamp et al., 2022; Li et al., 2023; Ouyang et al., 2022; Wei et al., 2023) have been trained on code, enabling them to assist with programming tasks and automate software development. While LLMs have shown success in generating isolated function-level Python code (Chen et al., 2021; Austin et al., 2021), they face challenges when generating system-level C++ code in large repositories due to complex APIs and broader contextual dependencies (Du et al., 2023; Wang et al., 2024). Thus, adopting LLMs to generate code in real-world software remains difficult (Jimenez et al.). Our work addresses this by developing a domain-specific language (DSL) that provides a high-level abstraction to help LLMs generate mapper code more effectively.

**LLM Optimization with Feedback** More recently, LLMs have been used to solve optimization problems. Usually, optimization problems exist in numerical domains (Boyd & Vandenberghe, 2004). AhmadiTeshnizi et al. (2023) used LLMs to solve structured optimization problems, such as mixed-integer linear programming (LP). Yang et al. (2024) used LLM to solve unstructured problems as black-box optimization. Follow-up work by Nie et al. (2024) showed the effectiveness of feedback in finding the global minima of black-box functions. Building on such ideas, Cheng et al. (2024) formally defined a class of optimization problems solvable by LLMs and proposed an optimizer based on the execution graph of a program. Later, Yuksekgonul et al. (2024) proposed an LLM optimizer inspired by gradient descent. These recent developments provide an opportunity to use LLMs to optimize code not for task completion but for improving over a performance metric.

## 3 MAPPER GENERATION TASK

**Task Definition** The problem we address is the automated generation of high-performance mappers for the Legion parallel programming framework (Bauer et al., 2012). Mappers determine how tasks are assigned to processors and how data is placed in memory to optimize performance. A well-designed mapper can achieve up to $10\times$ speedup compared to random mapping strategies. However, mappers are highly context-specific and must be carefully tailored to an application's input and the machine's architecture. Finding an optimal mapper is akin to solving a combinatorial optimization problem—a process that is time-consuming, labor-intensive, and traditionally performed by experienced performance engineers. The search space for discovering the best mapping strategy is vast, growing to $2^{14}$ even for the simplest scientific applications (as shown in prior work with a smaller search space (SFX Teixeira et al., 2023) than ours), making it challenging to efficiently explore all possible solutions. Moreover, even with a clear mapping strategy, writing the corresponding mapper requires experts to produce hundreds of lines of low-level C++ code, a task that can take several hours. As a result, the entire performance tuning process can take several days to complete.

**Mapping Decisions** Now we elaborate on the decisions that a mapper has to make. The first critical decision is the *processor selection* for each task, determining whether a task is better suited for GPUs, CPUs, or OpenMP runtime. This choice depends on factors such as task size, GPU memory, and kernel launch latency. For example, tiny tasks that require very little computation may prefer to run on CPUs due to the GPU kernel launch overhead, whereas tasks with large memory footprints might prefer to be assigned to OpenMP or CPU when GPU memory is insufficient.

Next, the *memory placement* of data across different memory spaces is crucial to performance. A mapper must decide where to place data — in the GPU's FrameBuffer for faster access, in ZeroCopy

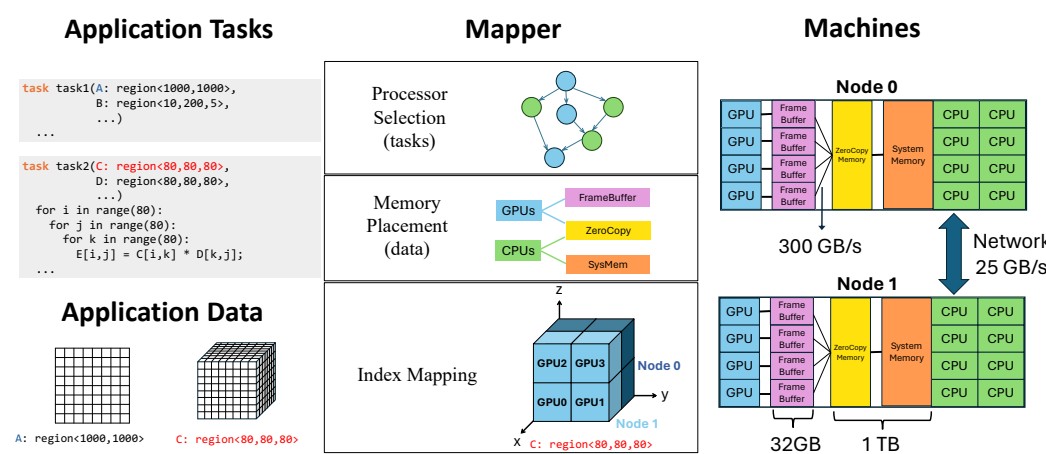

Figure 2: Mappers decide the placement of each task in the task graph to processors, the placement of data to memory, and how the iteration space of data is partitioned and mapped to different processors.

memory for shared access between CPU and GPU, or in CPU system memory for large data. Each choice introduces a trade-off between memory access speed, memory usage, and transfer overhead.

The *memory layout* option determines the optimal memory arrangement for data structures. Depending on the memory access patterns and the underlying hardware, selecting between a Struct of Arrays (SOA) or an Array of Structures (AOS) layout, along with constraints on array ordering (Fortran-order or C-order) and memory alignment, can significantly impact performance due to cache efficiency and data locality. Such choice is task-dependent and processor-dependent.

Finally, mappers need to decide how to perform *index mapping*. As shown in Figure 2, index mapping controls how the data (or more accurately, parallel for loops of task launches) is partitioned and mapped to the processors. The data (or tensors) can be multi-dimensional, and the distributed machines can also be viewed as a processor space (nodes and processors within the node). Index mapping controls the mapping between the two index spaces. Prior work Unger et al. (2022); Zheng et al. (2022) shows that searching over how data is partitioned and mapped can significantly change inter-processor communication volume, which affects performance. Commonly-used index mapping functions are shown in Appendix A.5.

## 4 A New Approach: Generation and Optimization in a DSL

### 4.1 Domain-Specific Language Design

Figure 3a presents an example DSL mapper that illustrates the key features of our domain-specific mapping language. By contrast, Figure 3b shows a (already simplified) code snippet extracted from a C++ mapper. Notably, the C++ snippet covers only part of the functionality that the IndexMap statement (which we discuss in detail below) in the DSL provides. While a typical C++ mapper may require around 400 lines of code, the equivalent DSL mapper can achieve the same outcome in under 30 lines — a tenfold reduction. This significant decrease in code complexity makes the DSL more approachable as the code generation target for LLMs, as it abstracts away the low-level details.

Next, we will explain the DSL's design, highlighting the immense search space it opens for optimization, which makes generating high-performance mappers a challenging task.

**The Task statement** (Line: 2) performs processor selection for tasks. Although it may seem straightforward to always prefer GPUs, this decision is complicated by other factors such as limited GPU memory and kernel execution time as explained in Section 3. This is a per-task decision.

**The Region statement** (Line: 5) performs memory placement for data. Possible choices include GPU's FrameBuffer or ZeroCopy memory, CPU's System memory or RDMA memory. Such de-

```
1  # Map task0 to GPU.
2  Task task0 GPU;
3
4  # Place certain data onto GPU ZeroCopy.
5  Region * ghost_region GPU ZCMEM
6
7  # Specify layout in memory
8  # (aligned to 64 bytes)
9  Layout * * * C_order SOA Align==64
10
11 # Define a cyclic mapping strategy
12 def cyclic(Task task):
13     ip = task.ipoint;
14     mgpu = Machine(GPU);
15     node_idx = ip[0] % mgpu.size[0];
16     gpu_idx  = ip[0] % mgpu.size[1];
17     return mgpu[node_idx, gpu_idx];
18
19 IndexTaskMap task4 cyclic
```

```
1  void slice_task(const Task& task,
2                  const SliceTaskInput &input,
3                  SliceTaskOutput &output) {
4    vector<Processor> targets =
5      this->select_targets_for_task(ctx, task);
6    DomainT<2> space = input.domain;
7    Point<2> num_points =
8        space.bounds.hi - space.bounds.lo + ones;
9    Rect<2> blocks(zeroes, num_blocks - ones);
10   ...// **126 lines of C++ code ommitted**
11   for (PointInRectIterator<2> it(blocks); it() !=
12       NULL; it++) {
13     DomainT<2,coord_t> slice_space;
14     TaskSlice slice;
15     slice.domain = {slice_lo, slice_hi};
16     slice.proc = targets[index++ % targets.size()];
17     output.slices.push_back(slice);
18   }
19 }
```

(a) An example mapper in DSL      (b) Simplified code snippet from a C++ mapper

Figure 3: A DSL mapper and a short code snippet from C++ mapper. The C++ mapper has already been simplified, and the shown C++ snippet only contributes to part of the functionality achieved by the IndexTaskMap statement on Line 19 in the DSL mapper.

cisions need to be made for each argument of each task, which opens up a huge trade-off space between memory usage, task execution time, and data transfer costs as explained in Section 3.

**The Layout statement** (Line 9) defines memory layouts, supporting both SOA (Struct of Arrays) and AOS (Array of Structures), as well as array ordering constraints like C_order, F_order, and memory alignment. Optimizing memory layout is crucial for performance, as access patterns vary with tasks, processors, and data structures. This is a per-task, per-data, per-processor decision.

**The IndexMap statement** (Line 19) enables the mapping of tasks to processors using custom functions, such as cyclic or block mapping strategies. This creates a mapping between the index space defined in application code (e.g., for loops) and the processor space of distributed machines. Many parallel operators (e.g., matrix multiplications) are based on such loops, and deciding how to map them to distributed machines is inherently difficult. The DSL enables users to express arbitrary arithmetic mappings between two index spaces, significantly increasing the complexity of the search space. Additionally, we introduce primitives that transform the processor space, allowing for more flexible and precise index mapping decisions, as detailed in Appendix A.4.

We distill the key performance-critical aspects of mapping into language constructs that are easily expressed in the DSL. To implement our DSL, we develop a compiler that can translate the mapper written in DSL into low-level C++ mapping APIs. By providing a higher-level abstraction than C++ APIs, the DSL simplifies interfacing with LLMs, allowing them to efficiently address the complex optimization challenges of generating high-performance mappers. This optimization process, including the role of LLMs in automating these decisions, is detailed in Section 4.2.

## 4.2 LEARNING TO GENERATE VIA INTERACTIVE FEEDBACK

We conceptualize the mapper generation problem as a search for valid DSL programs. Since effective mappers must account for both specific application inputs and the underlying hardware, this task is well-suited to reinforcement learning (RL).

Our method combines the code-generation strengths of LLMs with an agent-based framework built on Trace (Cheng et al., 2024). In this framework, LLMs act as optimizers, iteratively generating mappers in the DSL and refining them based on real-time feedback. This feedback loop dramatically reduces the time required to produce high-quality mappers, from days to minutes, making the RL-driven agent particularly effective in this context-sensitive task.

Figure 4a illustrates the code templates used to construct the self-adapting agent with Trace. These templates guide the LLM in generating syntactically correct DSL code, define the search space, and provide heuristics for mapping decisions. Their design is critical to the quality of the generated code, ensuring the LLM starts with valid DSL syntax, a reasonable initial strategy, and an understanding of the performance impact of each decision (detailed in the docstring). Functions

```
1 class MapperGenerator(trace.Module):
2   def forward(self, app):
3     task_stmt   = self.gen_task_stmt(app)
4     region_stmt = self.gen_region_stmt(app)
5     layout_stmt = self.gen_layout_stmt(app)
6     ...
7     return task_stmt + region_stmt + ...
8
9   @trace.bundle(trainable=True)
10  def gen_task_stmt(self, app) -> str:
11    """
12    Generate the policy for placing tasks.
13    Example generated code:
14    Task * GPU; Task task1 CPU;
15    """
16    code = ""
17    for task in app.tasks:
18        proc = random.choice(["GPU","CPU"])
19        code += f'Task {task} {proc};\n'
20    return code
```

```
1 policy = MapperGenerator()
2 params = policy.parameters()
3 optimizer = trace.Optimizer(params)
4
5 app  = GetApplicationInfo()
6 test = GetMapperEvaluator(app)
7
8 for i in range(iterations):
9   # Forward pass
10  try:
11    mapper = policy(app)
12    # feedback (str) contains performance
13    feedback = test(mapper)
14  except TraceExecutionError as e:
15    feedback = str(e)
16    target = e.exception_node
17  # Backward pass and update
18  optimizer.zero_feedback()
19  optimizer.backward(target, feedback)
20  optimizer.step()
```

(a) The decision procedures in generating a mapper.    (b) Trainable policy using Trace operators.

Figure 4: We build a self-adapting agent with Trace. We need to provide the doc-string to explain each mapping decision, and initial heuristics to indicate the optimization space and the DSL syntax (Figure 4a). Then we use the Trace optimizer, which is similar to PyTorch (Figure 4b).

responsible for decision-making are annotated with `trainable=True`, allowing them to evolve iteratively as the LLM refines its strategy. All functions marked as trainable (along with the function body and docstring) are included in the LLM prompt to support continuous improvement throughout the optimization process. We show the full template in Section A.10.

In Figure 4b, we set up the agent using the optimizer from the Trace framework. We first retrieve the application's information and initialize the testing environment. During each iteration, the current policy generates a mapper, which is then evaluated to provide feedback. The feedback can indicate a failure in trace execution (if the trainable functions are not properly executed), an execution failure (e.g., running out of GPU memory), or performance metrics (if the mapper runs successfully). Furthermore, we provide additional feedback that can further guide LLMs by 1) providing the error explanation if there is an execution failure (i.e., a more informative failure message); and 2) providing suggestions on how to change the mapping decision when errors happen. We run an ablation study in Section 5.4 in our experiments to demonstrate the effectiveness of the additional feedback.

## 5 EVALUATION

Experiments are conducted on a GPU cluster where each node has two Intel 10-core E5-2640 v4 CPUs, 256G main memory, and four NVIDIA Tesla P100 GPUs. Regarding the LLM, we use gpt-4o-2024-08-06.

### 5.1 EFFECTIVENESS OF THE DSL

To evaluate the necessity and effectiveness of our DSL in mapper code generation, we compared the generation of mappers in C++ versus DSL. We devised 10 mapping strategies, each described in natural language, to serve as test cases for code generation.[1] A complete list of these strategies is provided in Appendix A.11. It is important to note that the objective is not to optimize the performance of any specific application, but rather to assess whether the LLM can accurately generate a C++/DSL mapper based on a given strategy.

We provided the same types of materials in the prompt to ensure a fair comparison. We provided documentation[2], example mapper programs, and starting code for both DSL and C++ interfaces. We measured the success rate of generating mappers that can pass compilation and pass test cases.

---

[1]For example, one strategy is "aligning all data to 64 bytes in memory and utilizing Fortran ordering for multi-dimensional data."

[2]We created documentation covering all features of DSL. We used existing documentation for C++ mapper.

| Code Generation Target | Mapping Strategy | | | | | | | | | | Success Rate |
|---|---|---|---|---|---|---|---|---|---|---|---|
| | 1 | 2 | 3 | 4 | 5 | 6 | 7 | 8 | 9 | 10 | |
| C++ (single trial) | ✗ | – | – | ✗ | – | – | ✗ | ✗ | – | – | 0% |
| C++ (iterative refine) | ✗ | – | – | ✗ | ✗ | ✗ | ✗ | ✗ | ✗ | ✗ | 0% |
| DSL (single trial) | ✓ | ✓ | ✓ | ✓ | ✓ | – | ✓ | ✓ | ✓ | – | 80% |

Table 1: **Compilation and Strategy Test**. We show the success rate for code generation given 10 mapping strategies. Generating DSL code significantly outperforms generating C++ (either without or with compiler feedback). – fails to compile, ✗ compiles but fails the test, and ✓ passes the test.

For C++ code generation, we enhanced the process by incorporating compiler feedback. The LLM received compiler error messages and iteratively refined the code, with up to 10 iterations allowed. We use DSPy (Khattab et al., 2023) to build our interface.

Table 1 presents the performance of the LLM in code generation in three settings: C++ single trial (without compiler feedback), C++ with compiler feedback, and DSL. We evaluate the LLM with 10 different mapping strategies. As observed, the DSL approach consistently outperforms the other settings. These results confirm that the LLM is poor at generating system-level C++ code and highlights the effectiveness and necessity of our DSL design in significantly enhancing the code generation capabilities for mapping.

**Failure Case Analysis** The compilation errors in C++ mapper generation arise because LLMs are unable to manage the framework-specific contextual dependencies inherent to low-level system software programming. For instance, LLMs generate variable names or references that do not exist within the provided codebase. The reason is that the documentation and example code are so long and complex that LLMs fail to retrieve the related information and hallucinate the variable names.

In the cases where the code compiles but fails the test cases, the failures are exclusively in C++. The root cause lies in the complexity of implementing a C++ mapper, which requires a deep understanding of the API documentation and reasoning about the code examples. As outlined in the documentation, multiple APIs must work in concert to accomplish specific tasks, yet LLMs are unable to grasp this level of coordination. For instance, to implement the index mapping feature, LLMs need to override several different functions together (e.g., one function deciding the target node index and the other deciding the target processor index, and some other functions to inform the runtime that the heuristics has been changed), which is too challenging for LLMs. In contrast, this has been simplified to just one simple function in the DSL (as shown in Figure 3).

While compiler feedback can guide LLMs in avoiding trivial errors (e.g., such as generating non-existent variable names), it cannot bridge the gap in understanding the intricacies of low-level C++ mapping APIs. This limitation explains why all attempts at C++ code generation ultimately fail.

In contrast, only two test cases fail in the DSL context on the single trial, both due to compilation errors stemming from incorrect syntax. This is understandable given that the DSL is a newly designed language with limited training data available for the LLM. Thus, for performance optimization experiments, we do not attempt to have LLMs optimize over C++ mappers, as results sufficiently demonstrate that LLMs struggle to generate C++ code that meets the specifications of what a desired mapper should do, let alone explore a large search space of different mapping strategies.

## 5.2 ACCELERATING SCIENTIFIC APPLICATIONS

In this experiment, we compared the performance of different mappers: expert-written mappers, randomly generated mappers, and mappers generated by LLM agents. All mappers are implemented using our DSL. Our goal is to determine whether LLMs can effectively explore the search space of mappers by generating high-performance DSL mappers. We do not attempt to have LLMs search over C++ mappers, as results from Section 5.1 sufficiently demonstrate that LLMs struggle to generate C++ code that meets a separate specification of what the desired mapper should do, let alone explore the large search space of different mapping strategies.

The expert-written mappers were manually developed and optimized by domain experts as part of the application development process. We re-implemented these expert-written C++ mappers using our DSL to establish a ground truth for comparison with our approach. Validation confirmed that

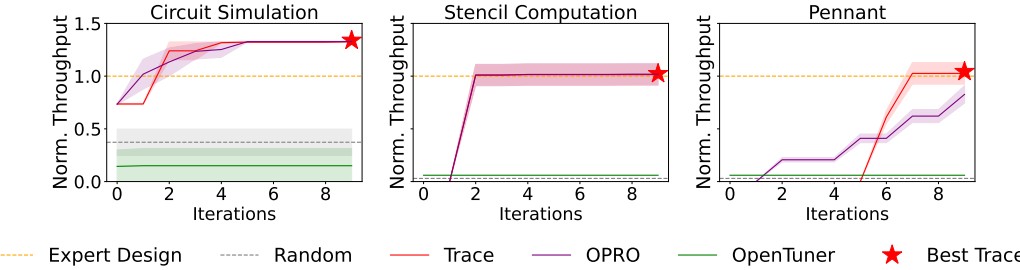

Figure 5: Normalized throughput for scientific applications comparing expert-written mappers, random mappers, best mappers found by Trace-OptoPrime, and the average optimization trajectories of Trace-OptoPrime, Trace-OPRO, and OpenTuner in 10 iterations across 5 runs.

the DSL-based mappers achieve performance equivalent to the original C++ mappers, providing a fair basis for evaluating the performance of our generated mappers.

The randomly generated mappers, which makes random mapping decisions, serve as a baseline for this experiment. We ran 10 random mappers and reported the average performance.

For LLM-generated mappers, we implemented the search using Trace (Cheng et al., 2024), which employs LLMs as optimizers. We tested both the OptoPrime and OPRO (Yang et al.) search algorithms, running 10 iterations for each application. Due to the stochastic nature of LLM outputs, we repeated the optimization process 5 times and averaged the results. We also report the best mapper found by Trace-OptoPrime across the 5 runs.

We use the following three scientific applications as our benchmarks. The circuit simulation benchmark (Bauer et al., 2012) models the behavior of an electrical circuit by simulating currents and voltages across interconnected nodes and wires. The stencil computation benchmark (Van der Wijngaart & Mattson, 2014) simulates a 2D grid where each point's value is updated based on its neighbors using a stencil pattern. The Pennant benchmark (Ferenbaugh, 2015) models unstructured mesh, Lagrangian staggered-grid hydrodynamics, used for simulating compressible flow.

The performance-critical mapping decisions for these applications are the processor type selection, memory placement of data, and the layout selection for data. Other choices available to mappers do not change the performance of these applications much. The simplest application above (with the smallest search space) is Stencil, which contains 2 tasks and 12 data collection arguments. Each task and data argument has two choices of placement, together with additional 4 layout choices for each data, forming an optimization space of $2^{38}$.

**Results** We use normalized throughput as the performance metric in Figure 5, where higher values indicate better performance. The throughput is normalized w.r.t the expert-written mappers. As shown in Figure 5, random mappers are consistently the least effective across all applications, highlighting the importance of mapping decisions on application performance. When comparing the optimization trajectories of Trace-OptoPrime and Trace-OPRO, Trace-OptoPrime performs similarly to Trace-OPRO, and is slightly better than Trace-OPRO on Pennant. All the best mappers found by Trace-OptoPrime can at least match the performance of expert mappers.

Interestingly, the best mapper identified in the Circuit benchmark outperforms the expert mapper by 34%. Upon manual investigation, we observed that the key difference lies in memory placement: the best mapper allocates two data collections to the GPU FrameBuffer memory, whereas the expert mapper places these collections in GPU ZeroCopy memory. This strategy reduces task execution time, despite a slight increase in inter-GPU communication costs, ultimately leading to improved overall performance. For the Pennant benchmark, while there is a minor difference in data collection placement, the final performance results between the two mappers are nearly equivalent.

For each application, the search completes within 10 minutes, significantly reducing mapper development time from days to minutes. This substantial improvement highlights the efficiency of our LLM-enhanced DSL in quickly generating high-performance mappers, offering clear benefits for both developers and application performance.

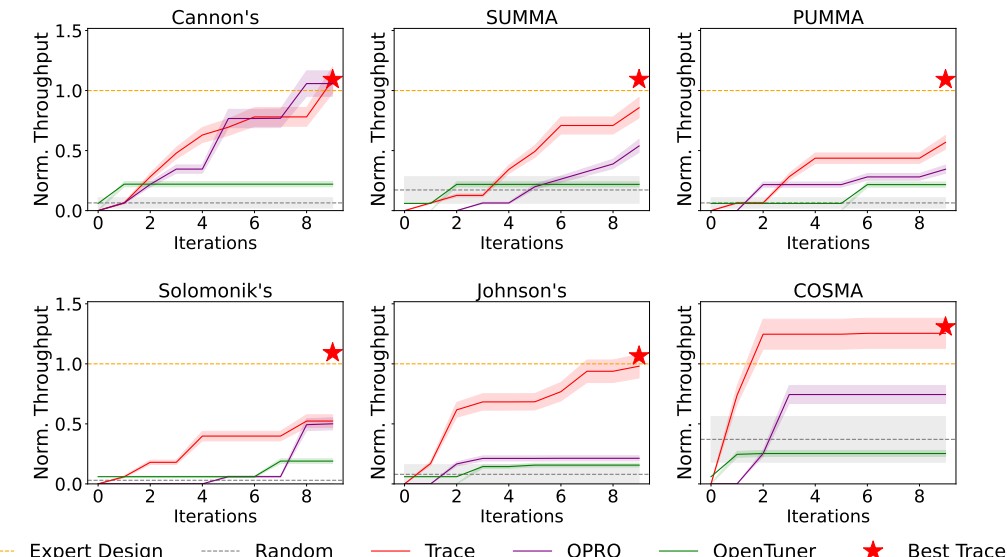

Figure 6: Normalized throughput for matrix-multiplication algorithms. We compare expert-written mappers, random mappers, best mappers found by Trace-OptoPrime, and the average optimization trajectories of Trace-OptoPrime, Trace-OPRO, and OpenTuner in 10 iterations across 5 runs.

## 5.3 ACCELERATING MATRIX MULTIPLICATIONS

We follow the same experimental setup as described in Section 5.2, with the key difference being the focus on matrix multiplication algorithms in this subsection. Unlike the applications in Section 5.2, where processor selection and memory placement are the critical performance factors, the performance-critical mapping decision here is *index mapping*, which is the choice of how to launch concurrent tasks and distribute the tiles of matrices across multiple GPUs.

In the DSL mapper, achieving an effective index mapping requires specifying a function that maps the iteration space (defined by the parallel for loop) to the processor space of the machine (please see the IndexMap statement discussed in Section 4.1). Based on our estimates, each algorithm has approximately $10^9$ possible choices for index mapping. We show some of the mapping functions in Appendix A.8

We target 6 different matrix multiplication algorithms: Cannon's (Cannon, 1969), SUMMA (Van De Geijn & Watts, 1997), PUMMA (Choi et al., 1994), Johnson's (Agarwal et al., 1995), Solomonik's (Solomonik & Demmel, 2011), and COSMA (Kwasniewski et al., 2019). Each algorithm exhibits different performance characteristics and may be the preferred implementation of matrix multiply depending on the target machine, input size, and mapping decisions. Our goal is to explore and identify better mappings for these algorithms. We categorize and elaborate further on these algorithms in Appendix A.7.

**Results** As shown in Figure 6, the expert mappers' compute throughput is normalized to 1.0, reflecting the mapping decisions specified by the algorithms. Random mappers yield the lowest throughput, underscoring the critical role of well-designed mappers. The best mappers found by Trace-OptoPrime consistently outperform the expert mappers, achieving speedups ranging from 9% to 31%. For both PUMMA and Solomonik's, the throughput of the best mapper discovered by Trace-OptoPrime is significantly higher than the average optimization trajectory across 5 runs. This variability is due to the inherent randomness of LLMs in our experiments, where even subtle changes in mappers can lead to notable performance differences. When comparing optimization trajectories of Trace-OptoPrime and Trace-OPRO, Trace-OptoPrime shows significant performance gains in SUMMA, PUMMA, Johnson's, and COSMA, while achieving similar results on other benchmarks.

Finally, we inspected and compared the mapping decisions of the expert mappers with those of the best mappers found through our search. The performance improvements are entirely attributable to

more effective index mapping. Index mapping governs the partitioning and distribution of data, in this case, matrices, across the GPUs. The optimized mapping reduces inter-GPU communication and enhances data locality, leading to improved performance in parallel matrix multiplication algorithms.

## 5.4 ABLATION STUDY OF FEEDBACK

The quality of the feedback directly affects the success of the optimization process. We evaluate three types of feedback to assess how different feedback influences the performance of LLM-based optimizers during mapper search: (1) system feedback (compile error, execution error, or performance metrics), (2) error explanations, and (3) suggestions for mapper adjustments.

The first feedback message includes only system feedback (labeled **System** in Figure 7). Next, we evaluate an enhanced feedback message that includes both system feedback and error explanations (labeled **System+Explain**). Finally, we provide the full feedback message, including all three types (labeled **System+Explain+Suggest**), corresponding to the Trace-OptoPrime results shown in Figure 5 and Figure 6. We evaluate three benchmarks, a subset of the total 9 applications.

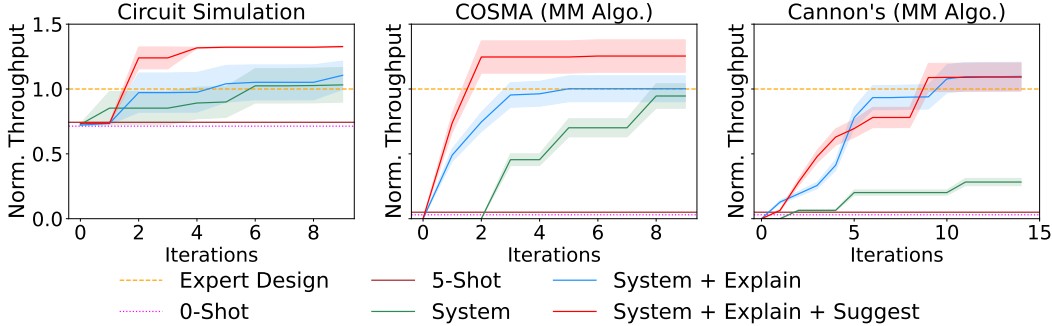

Figure 7: Comparison of three types of feedback design. **0-Shot** and **5-Shot** have no feedback. **System** provides only the system execution information (performance numbers or error messages) to the optimizer. **Explain** provides additional explanations of execution errors. **Suggest** provides modification proposals to the mappers. All feedback is automatically generated.

As shown in Figure 7, across all three benchmarks, the full feedback message consistently achieves the highest throughput after 10 iterations, followed by the one without guidance. The system-only feedback performs the worst among the three. While the degree of impact from the feedback types varies across benchmarks, the results clearly demonstrate the critical role of high-quality feedback in guiding LLM optimizers to discover more efficient mappers.

## 6 CONCLUSION

In this paper, we addressed the challenges of automating mapper generation in task-based programming through the use of LLMs and a Domain-Specific Language (DSL). By designing a high-level DSL, we effectively simplified the complex task of generating low-level C++ mappers, enabling LLMs to handle mapper generation without deep knowledge of system intricacies. Additionally, we formulated the mapper generation task as a discrete optimization problem, leveraging reinforcement learning (RL) techniques to explore a structured and constrained search space defined by the DSL.

Our experimental results demonstrate the efficacy of this approach, with LLM-generated mappers achieving up to 34% speedup over expert-written mappers across 9 benchmarks. For matrix multiplication algorithms, we observed a performance boost of up to 31%. These findings show that our LLM-enhanced DSL significantly reduces development time from days to minutes, benefiting both human developers and the performance of parallel applications.

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

# A APPENDIX

## A.1 LINES OF CODE COMPARISON BETWEEN DSL AND C++

| Application | 1 | 2 | 3 | 4 | 5 | 6 | 7 | 8 | 9 | Avg. |
|---|---|---|---|---|---|---|---|---|---|---|
| LoC in C++ | 347 | 306 | 379 | 447 | 437 | 430 | 428 | 433 | 448 | 406 |
| LoC in DSL | 16 | 14 | 16 | 38 | 38 | 38 | 33 | 38 | 32 | 29 |
| LoC Reduction | 22× | 22× | 24× | 12× | 12× | 11× | 13× | 11× | 14× | 14× |

Table A1: Lines of Code (LoC) of mapper written in DSL, and LoC reduction compared with C++

## A.2 DSL GRAMMAR

We show the DSL grammar in Figure A1.

$$
\begin{array}{rcl}
\texttt{Program} & ::= & \texttt{Statement}^+ \\
\texttt{Statement} & ::= & \texttt{IndexTaskMap } \textbf{TaskName var} \mid \\
& & \texttt{SingleTaskMap } \textbf{TaskName var} \mid \\
& & \texttt{FuncDef} \mid \texttt{TaskMapModifier} \\
& & \texttt{RegionMapping} \mid \texttt{DataLayout} \\
\\
\texttt{Proc} & ::= & \texttt{CPU} \mid \texttt{GPU} \mid \ldots \\
\texttt{Memory} & ::= & \texttt{SYSMEM} \mid \texttt{FBMEM} \mid \texttt{ZCMEM} \mid \ldots \\
\\
\texttt{TaskMapModifier} & ::= & \texttt{GarbageCollect } \textbf{TaskName RegionName} \mid \\
& & \texttt{Backpressure } \textbf{TaskName int} \mid \ldots \\
\texttt{RegionMapping} & ::= & \texttt{Region } \textbf{TaskName RegionName } \texttt{Proc Memory}^+ \\
\\
\texttt{DataLayout} & ::= & \texttt{Layout } \textbf{TaskName RegionName } \texttt{Proc Constraint}^+ \\
\texttt{Constraint} & ::= & \texttt{SOA} \mid \texttt{AOS} \mid \texttt{C\_order} \mid \texttt{F\_order} \mid \texttt{Align } \mathbf{==} \textbf{ int} \\
\\
\texttt{FuncDef} & ::= & \texttt{def } \textbf{var}(\textbf{var}^+): \texttt{FuncStmt}^+ \\
\texttt{FuncStmt} & ::= & \textbf{var} = \texttt{Expr} \mid \texttt{return Expr} \\
\texttt{Expr} & ::= & \textbf{var} \mid \textbf{var}(\texttt{Expr}^+) \mid \texttt{Machine(Proc)} \mid \texttt{Expr.Expr} \mid \\
& & \texttt{Expr } \textbf{Op} \texttt{ Expr} \mid \texttt{(Expr)} \mid \texttt{Expr[Expr]} \mid * \texttt{ Expr} \\
& & \texttt{Expr ? Expr : Expr}
\end{array}
$$

Figure A1: Grammar of DSL. A DSL program is a list of statements, each of which controls one aspect of mapping.

## A.3 IMPLEMENTATION OF THE TRANSLATION

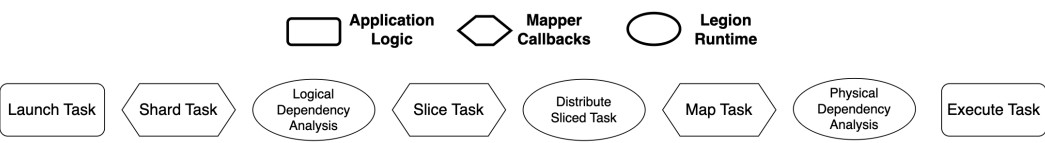

Figure A2: Pipeline stages before task execution in Legion

We outline how our implementation translates our DSL into Legion's low-level C++ mapping interface. The Legion runtime (and other modern tasking runtimes) employs a pipelined execution model, where multiple tasks are going through different stages of analysis and execution concurrently. Each pipeline stage handles a different portion of a task's analysis and execution, and some pipeline stages may interact with the user's mapper through the mapping interface. Since many tasks progress through the pipeline simultaneously, callbacks are often interleaved with callbacks for other tasks.

Legion is a distributed runtime—multiple *ranks* of the runtime run in different parts of the machine. Each instance of the runtime exclusively manages some subset of the machine's resources.

Figure A2 shows a simplified version of Legion's task pipeline. We are not concerned with the details of what the stages do; we focus on the parts relevant to the mapping callbacks. We focus on the translation of `IndexTaskMap` and `Region` statements, as these illustrate all of the interesting issues.

When the application launches an index task, the local rank first invokes the `shard` function of the user's mapper to choose the ranks where subsets of the index launch should be sent to complete the rest of the mapping process. Some tasks may remain in the pipeline on the local rank, while other tasks are sent to other ranks for mapping. Regardless of where the tasks are sent, the subsets of tasks sent to each rank are themselves still represented as an index launch.

After the sharded index task launches arrive at the ranks where they will finish mapping, the callback `slice_task` is invoked. The `slice_task` function maps the tasks in the index launch onto processors, typically (but not necessarily) the processors of the rank that the (subset of the) index launch has been sharded to. We translate `IndexTaskMap` commands into the needed `shard_task` and `slice_task` callbacks, ensuring the two are properly coordinated and any concurrency issues are avoided. The key to this translation is that all of the machine transformations are invertible, which makes it possible to take a processor selected in a transformed machine model and map it back (possibly through multiple layers of machine transformations) to the corresponding processor in the physical machine.

After `slice_task` finishes, tasks in the pipeline are all individual point tasks. The distribution stage transfers any tasks that have been mapped to a processor owned by a different rank to their final destination. Once on the rank with the task's assigned processor, the mapper callback `map_task` decides how to map the regions, i.e., in which memory a physical instance of the region will be placed and with what layout. The compiler automatically generates the logic for `map_task` from the region and layout statements.

## A.4 PROCESSOR SPACE TRANSFORMATION FOR INDEX MAPPING

We define the semantics for each of DSL's transformation primitives in Figure A3. A transformation primitive is a function of the processor space $m$ that returns a transformed processor space $m'$, where $m$ and $m'$ are related through the mapping shown in the right-hand side of Figure A3.

Our transformations are inspired by widely used operations for changing the dimensionality of arrays in libraries such as NumPy, but the application to mapping is quite different. We now explain each transformation in detail.

The *split* transformation takes two arguments: an integer $i$ that indicates the dimension to be split, and the splitting factor $d$. Suppose $m$ is a processor space of size $(8, 8)$, then after executing $m' = m.\mathtt{split}(0, 2)$, $m'$ will be a processor space of size $(2, 4, 8)$. An important property of split and all DSL transformations is that they are invertible. Thus, mappers can work with the transformed space $m'$ but DSL can translate such uses back into the original processor space $m$ to identify which concrete processors to use. In this example, $m'[j_0, j_1, j_2] = m[j_0 + j_1 \times 2, j_2]$.

The *merge* transformation takes two dimensions of the original processor to be fused as its input. Suppose $m'$ is a processor space of size $(2, 4, 8)$. After applying $m'' = m'.\mathtt{merge}(0, 1)$, $m''$ will be a processor space of size $(8, 8)$. The processor indexed by $m''[j_0, j_1]$ corresponds to $m'[j_0 \% 2, j_0/2, j_1]$.

Transformations can be chained together. Suppose $m$ is a 2D processor space, and we start with $m' = m.\mathtt{split}(0, d)$, followed by $m'' = m'.\mathtt{merge}(0, 1)$. The final processor space $m''$ is

| **Transformation** | **Semantics** |
|---|---|
| $m' = m.\texttt{split}(i, d)$ | $m'[a_0, \ldots, a_{n+1}] := m[b_0, \ldots, b_{n-1}, b_n]$ 
 $b_t = \begin{cases} a_t & t < i \\ a_i + a_{i+1} \cdot d & t = i \\ a_{t+1} & t > i \end{cases}$ |
| $m' = m.\texttt{merge}(p, q)$ 
 $p < q$ | $m'[a_0, \ldots, a_{n-1}] := m[b_0, \ldots, b_{n-1}, b_n]$ 
 $b_t = \begin{cases} a_t & t < p \vee p < t < q \\ a_p \% m.size[p] & t = p \\ a_p / m.size[p] & t = q \\ a_{p-1} & t > q \end{cases}$ |
| $m' = m.\texttt{swap}(p, q)$ | $m'[a_0, \ldots, a_{n-1}] := m[b_0, \ldots, b_{n-1}]$ 
 $b_t = \begin{cases} a_q & t = p \\ a_p & t = q \\ a_t & t \neq p \wedge t \neq q \end{cases}$ |
| $m' = m.\texttt{slice}(i, low, high)$ 
 $0 \leq low \leq high < m.size[i]$ | $m'[a_0, \ldots, a_{n-1}] := m[b_0, \ldots, b_{n-1}]$ 
 $b_t = \begin{cases} a_i + low & t = i \\ a_t & t \neq i \end{cases}$ |

Figure A3: Semantics of processor space transformations expressed as mappings from the indices of the transformed processor space to the indices of the original processor space.

a 2D processor space. Now we will derive the index transformation from $m''$ to $m$ by applying the transformation rules from merge transformation and split transformation: $m''[j_0, j_1] = m'[j_0\%d, j_0/d, j_1] = m[(j_0\%d) + (j_0/d) \times d, j_1]$. The expression $(j_0\%d) + (j_0/d) \times d$ can be simplified to $j_0$ because the division operator $/$ between two integers rounds to zero. Therefore, $m''[j_0, j_1] = m[j_0, j_1]$, showing that the split and merge transformation primitives are inverses of each other.

The *swap* transformation primitive takes two parameters indicating the dimensions to be swapped and returns a processor space where the two indices of the chosen dimensions are flipped. The swap transformation is often combined with merge: The merge primitive linearizes two dimensions into one, but there is a choice whether to use row-major or column-major iteration order in the linearization. Users can change the iteration order for the merge by swapping the two dimensions.

The *slice* transformation primitive takes three parameters, the dimension to slice, and the lower bound and upper bound of the dimension. The index mapping rule for the slice transformation is to add a constant shift in the chosen dimension. The slice transformation primitive can be useful if users want to map the iteration space to only part of the original processor space. If two iteration spaces can be executed concurrently in the program, users can map one iteration space to half of the processors and map the other iteration space to the other half of the processors. In this case, the slice transformation allows users to map an iteration space to the selected portion of the processor space.

## A.5 COMMON INDEX MAPPING FUNCTIONS

| Distribution | Iteration Space | Processor Space | Transformation | Mapping Function |
|---|---|---|---|---|
| block2D | | | `m = Machine(GPU)` | ```def block2D(Tuple ipoint, Tuple ispace):`
`    idx = ipoint * m.size / ispace`
`    return m[*idx]``` |
| block1D_x | | | `m = Machine(GPU)`
`m1 = m.merge(0, 1).split(0, 1)` | ```def block1D_x(Tuple ipoint, Tuple ispace):`
`    idx = ipoint * m1.size / ispace`
`    return m1[*idx]``` |
| block1D_y | | | `m = Machine(GPU)`
`m2 = m.merge(0, 1).split(0, 4)` | ```def block1D_y(Tuple ipoint, Tuple ispace):`
`    idx = ipoint * m2.size / ispace`
`    return m2[*idx]``` |
| cyclic2D | | | `m = Machine(GPU)` | ```def cyclic2D(Tuple ipoint, Tuple ispace):`
`    idx = ipoint % m.size`
`    return m[*idx]``` |
| cyclic1D_x | | | `m = Machine(GPU)`
`m1 = m.merge(0, 1).split(0, 1)` | ```def cyclic1D_x(Tuple ipoint, Tuple ispace):`
`    idx = ipoint % m1.size`
`    return m1[idx]``` |
| cyclic1D_y | | | `m = Machine(GPU)`
`m2 = m.merge(0, 1).split(0, 4)` | ```def cyclic1D_y(Tuple ipoint, Tuple ispace):`
`    idx = ipoint % m2.size`
`    return m2[idx]``` |
| block-cyclic | | | `m = Machine(GPU)` | ```def blockcyclic(Tuple ipoint, Tuple ispace):`
`    idx = ipoint / m.size % m.size`
`    return m[*idx]``` |

Figure A4: Common transformations and index mapping functions. The shaded subarea of the iteration space will be mapped to the shaded processor in the processor space. The transformation code can transform the original $(2, 2)$ processor space into the desired processor space. The result processor space will be used for mapping in the user-defined function.

## A.6 EXAMPLES OF FEEDBACK CONFIGURATIONS

We give examples for the system feedback and enhanced feedback in Table A2. The enhanced feedback includes explanations of errors and suggestions for mapper modifications.

## A.7 PARALLEL MATRIX MULTIPLICATION ALGORITHMS

**2D Algorithms** Cannon's (Cannon, 1969) introduced a systolic communication pattern with tiled data partitioning for distributed matrix multiplication. PUMMA (Choi et al., 1994) and SUMMA (Van De Geijn & Watts, 1997) extended this approach by supporting non-square matrices and improving communication efficiency through pipelining. They are called 2D algorithms because they partition the matrices into 2D tiles and then map them onto the processor space.

**Non-2D Algorithms** Johnson's (Agarwal et al., 1995) introduced a 3D algorithm that partitions the input matrices into 3D tiles and uses additional memory per processor to reduce communication compared to 2D algorithms. Solomonik's (Solomonik & Demmel, 2011) balances between 2D and 3D approaches by using extra memory to further minimize communication. COSMA (Kwasniewski et al., 2019) takes a different approach by optimizing the processor grid and parallelization strategy based on the input size and the machine size.

## A.8 INDEX MAPPING FUNCTIONS USED BY MATRIX MULTIPLICATION ALGORITHMS

We show some index mapping functions used by matrix multiplication algorithms in Figure A5

## A.9 EXPLANATION OF INDEX MAPPING FOR SOLOMONIK'S ALGORITHM

Figure A6 shows a mapper for the Solomonik's algorithm on a 2-node machine with 4 GPUs per node. The result distribution for the 3D iteration space is that each node will get half of the whole iteration space by partitioning along the x-axis, and the 4 GPUs per node will perform a 2D block distribution over the y-z plane.

| Mapper | System Feedback | Enhanced Feedback | |
| --- | --- | --- | --- |
| | | **Explain** | **Suggest** |
| mapper1 | **Compile Error:** Syntax error, unexpected :, expecting { | N/A | There should be no colon : in function definition. |
| mapper2 | **Compile Error:** IndexTaskMap's function undefined | N/A | Define the IndexTaskMap function first before using it. |
| mapper3 | **Compile Error:** mgpu not found | N/A | Include `mgpu = Machine(GPU);` in the generated code. |
| mapper4 | **Execution Error:** Assertion failed: stride does not match expected value. | Memory layout is unexpected. | Adjust the layout constraints or move tasks to different processor types. |
| mapper5 | **Execution Error:** DGEMM parameter number 8 had an illegal value | Memory layout is unexpected. | Adjust the layout constraint. |
| mapper6 | **Execution Error:** Slice processor index out of bound | IndexTaskMap statements cause error. | Ensure that the first index of mgpu ends with `% mgpu.size[0]`, and the second element ends with `% mgpu.size[1]`. |
| mapper7 | **Execution Error:** Assertion 'event.exists()' failed | InstanceLimit statements cause error. | Avoid generating InstanceLimit statements. |
| mapper8 | **Performance Metric:** Execution time is 0.03s. | N/A | Move more tasks to GPU to reduce execution time. |
| mapper9 | **Performance Metric:** Achieved throughput = 4877 GFLOPS | N/A | Try using different `IndexTaskMap` or `SingleTaskMap` statements to maximize throughput. |

Table A2: System feedback and enhanced feedback (error explanations and adjustment suggestions) for different mappers.

| | |
|---|---|
| Helper functions,

Global variable | ```python
def block_primitive(Tuple ipoint, Tuple ispace, Tuple pspace, int dim1, int dim2):
    return ipoint[dim1] * pspace[dim2] / ispace[dim1]


def cyclic_primitive(Tuple ipoint, Tuple ispace, Tuple pspace, int dim1, int dim2):
    return ipoint[dim1] % pspace[dim2]


m_2d = Machine(GPU)
``` |
| Solomonik's
(function 1) | ```python
def hierarchical_block3D(Tuple ipoint, Tuple ispace):
    # split the 0th dimension into 3 dimensions
    m_4d = m_2d.decompose(0, ispace);
    # split the GPU dimension into 3 dimensions
    # sub iteration space for each node: ispace / m_4d[:-1]
    m_6d = m_4d.decompose(3, ispace / m_4d[:-1])
    upper = tuple(block_primitive(ipoint, ispace, m_6d, i, i) for i in (0,1,2))
    lower = tuple(cyclic_primitive(ipoint, ispace, m_6d, i, i + 3) for i in (0,1,2))
    return m_6d[*upper, *lower]
``` |
| Cannon's
PUMMA
SUMMA | ```python
def hierarchical_block2D(Tuple ipoint, Tuple ispace):
    # Similar to hierarchical_block3D except for the dimension of iteration space
    m_3d  = m_2d.decompose(0, ispace)
    m_4d  = m_3d.decompose(2, ispace / m_3d[:-1])
    upper = tuple(block_primitive(ipoint, ispace, m_4d, i, i) for i in (0, 1))
    lower = tuple(cyclic_primitive(ipoint, ispace, m_4d, i, i + 2) for i in (0, 1))
    return m_4d[*upper, *lower]
``` |
| Solomonik's
(function 2) | ```python
def linearize_cyclic(Tuple ipoint, Tuple ispace):
    linearized = ipoint[0] + ispace[0] * ipoint[1] + ispace[0] * ispace[1] * ipoint[2]
    # cyclic over node dimension and GPU dimension
    node_idx = linearized % m_2d.size[0]
    gpu_idx = (linearized / m_2d.size[0]) % m_2d.size[1]
    return m_2d[node_idx, gpu_idx]
``` |
| COSMA | ```python
def special_linearize3D(Tuple ipoint, Tuple ispace):
    # split the node dimension as equal as possible
    m_5d = m_2d.decompose(0, (1, 1, 1))
    gx = m_5d.size[2]
    gy = m_5d.size[1]
    linearized = ipoint[0] + ipoint[1] * gx + ipoint[2] * gx * gy
    return m_2d[linearized % m_2d.size[0], 0]
``` |
| Johnson's | ```python
def conditional_linearize3D(Tuple ipoint, Tuple ispace):
    grid_size = ispace[0] > ispace[2] ? ispace[0] : ispace[2]
    linearized = ipoint[0] + ipoint[1] * grid_size + ipoint[2] * grid_size * grid_size
    return m_2d[linearized % m_2d.size[0], 0]
``` |

Figure A5: Example mapping functions used by the mappers of matrix multiplication algorithms.

There is a dimension mismatch between the iteration space (3D) and the initial processor space (2D). To conduct the desired mapping required by the algorithm, we first apply the *split* transformation primitive four times (colored as red in the code). We apply the first (resp. last) two split transformations to make the node dimension (resp. GPU dimension) align with the 3D iteration space. We visualize the result 6D processor space as two 3D spaces. The first 3D space (representing the node dimension) is of size $(2, 1, 1)$ and the second 3D space (representing the GPU dimension) is of size $(1, 2, 2)$.

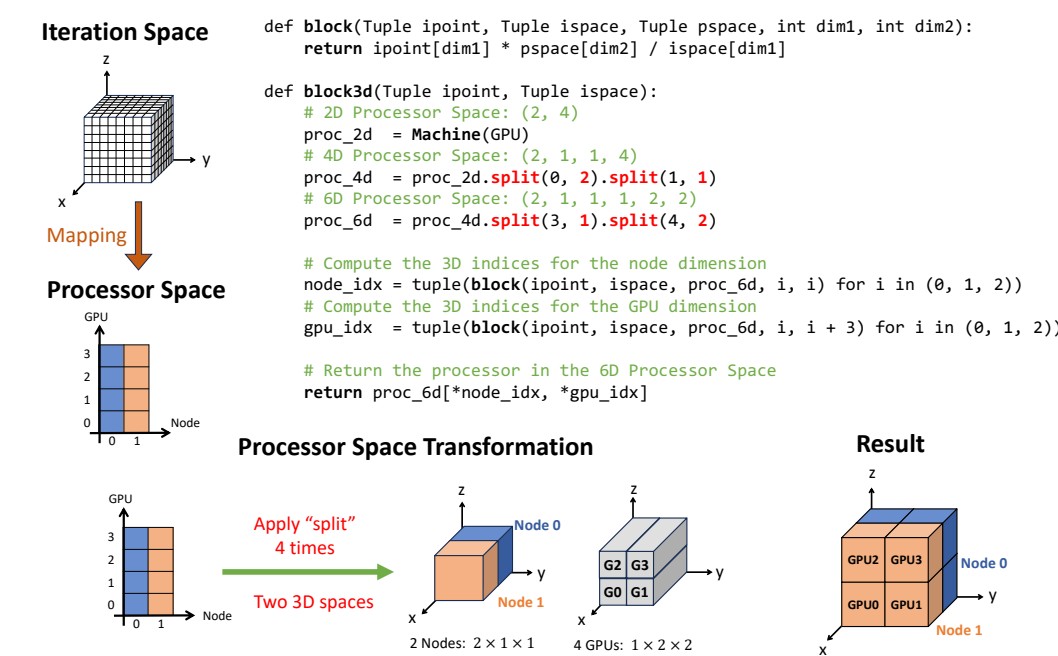

```
def block(Tuple ipoint, Tuple ispace, Tuple pspace, int dim1, int dim2):
    return ipoint[dim1] * pspace[dim2] / ispace[dim1]

def block3d(Tuple ipoint, Tuple ispace):
    # 2D Processor Space: (2, 4)
    proc_2d  = Machine(GPU)
    # 4D Processor Space: (2, 1, 1, 4)
    proc_4d  = proc_2d.split(0, 2).split(1, 1)
    # 6D Processor Space: (2, 1, 1, 1, 2, 2)
    proc_6d  = proc_4d.split(3, 1).split(4, 2)

    # Compute the 3D indices for the node dimension
    node_idx = tuple(block(ipoint, ispace, proc_6d, i, i) for i in (0, 1, 2))
    # Compute the 3D indices for the GPU dimension
    gpu_idx  = tuple(block(ipoint, ispace, proc_6d, i, i + 3) for i in (0, 1, 2))

    # Return the processor in the 6D Processor Space
    return proc_6d[*node_idx, *gpu_idx]
```

Figure A6: Mapper of the Solomonik's algorithm on a 2-node machine with 4 GPUs per node. The 2D processor space is transformed via the split transformation primitive into a 6D space (visualized as two 3D spaces).

### A.10  TRACE AGENT CODE

Trace (Cheng et al., 2024) uses Python decorators like @bundle to annotate Python programs. It allows us to design an LLM code generation agent as if we were writing a Python program ourselves. We first set up an end-to-end runnable Python program that can generate a valid mapper program by randomly making decisions over the search space. We show the high-level structure of our Trace Mapper below. At each optimization step, Trace will execute DSLMapperGenerator and collect the corresponding execution flow to build up a graph. Then it will make a call to an LLM to perform an update to any function that is decorated with @bundle(trainable=True). The DSLMapperGenerator is structured in the same way as providing a search space specified by the DSL, where an LLM optimizer can make decisions along the pre-designed axes. We note that this type of design is only enabled by recent developments like Trace and is much more challenging to do using older LLM-based frameworks.

```python
import opto.trace as trace

@trace.model
class DSLMapperGenerator():
    @trace.bundle(trainable=True)
    def task_decision(self, tasks):
        ...

    @trace.bundle(trainable=True)
    def region_decision(self, regions):
        ...

    @trace.bundle(trainable=True)
    def layout_decision(self):
        ...

    @trace.bundle(trainable=True)
    def instance_limit_decision(self, tasks):
        ...

    @trace.bundle(trainable=True)
    def index_task_map_decision(self, index_tasks):
```

```
24    @trace.bundle(trainable=True)
25    def single_task_map_decision(self, single_tasks):
26        ...
27
28    def generate_mapper(self):
29        """
30        Generate the final mapper code by combining all code statements.
31        """
32        task_statements = self.task_decision(self.tasks)
33        region_statements = self.region_decision(self.regions)
34        layout_statements = self.layout_decision()
35        instance_limit_statements = self.instance_limit_decision(self.tasks)
36        index_task_map_statements = self.index_task_map_decision(self.index_tasks,
      self.index_task_specification)
37        single_task_statements = self.single_task_map_decision(self.single_tasks)
38
39        code_statements = (
40            task_statements +
41            region_statements +
42            layout_statements +
43            instance_limit_statements +
44            index_task_map_statements +
45            single_task_statements
46        )
47        # Combine all code statements and function definitions into a single string
48        code_list = code_statements
49        mapper_code = str_join(node('\n'), *code_list)
50        return mapper_code
```

## A.11 MAPPING STRATEGIES

**Strategy 1:** Map the tasks of `calculate_new_currents`, `distribute_charge`, `update_voltages` onto GPUs in this way: linearize the 2D GPU processor space into 1D, then perform 1D block mapping from launch domain to the linearized 1D processor space.

```
1 Task * GPU,CPU; # for any task, run on GPU if supported
2 Region * *GPU FBMEM; # for any task, any region, if mapped onto GPU, use FBMEM as default
3 Region * * CPU SYSMEM; # if mapped onto CPU, use SYSMEM as default
4
5 Layout * * * SOA C_order;
6
7 mcpu = Machine(CPU);
8 mgpu = Machine(GPU);
9
10 ========== Above is fixed ==========
11 def linearblock(Task task) {
12     return mgpu[task.ipoint[0] / mgpu.size[1], task.ipoint[0] % mgpu.size[1]];
13 }
14
15 IndexTaskMap calculate_new_currents,distribute_charge,update_voltages linearblock;
```

**Strategy 2:** Place ghost/shared regions (rp_shared and rp_ghost) onto GPU zero-copy memory

```
1 Task * GPU,CPU; # for any task, run on GPU if supported
2
3 Region * * GPU FBMEM; # for any task, any region, if mapped onto GPU, use FBMEM as default
4 Region * * CPU SYSMEM; # if mapped onto CPU, use SYSMEM as default
5
6 Layout * * * SOA C_order;
7
8 mcpu = Machine(CPU);
9 mgpu = Machine(GPU);
10
11 ========== Above is fixed ==========
12
13 Region * rp_shared GPU ZCMEM;
14 Region * rp_ghost GPU ZCMEM;
```

**Strategy 3**: Use Array Of Struct (AOS) data layout for all data instead of the default SOA

```
1 Task * GPU,CPU; # for any task, run on GPU if supported
2
3 Region * * GPU FBMEM; # for any task, any region, if mapped onto GPU, use FBMEM as default
4 Region * * CPU SYSMEM; # if mapped onto CPU, use SYSMEM as default
5
6 mcpu = Machine(CPU);
```

```
 7 mgpu = Machine(GPU);
 8
 9 ========== Above is fixed ==========
10
11 Layout * * * AOS;
```

**Strategy 4**: Use Fortran ordering of data layout for all data instead of the default C order

```
 1 Task * GPU,CPU; # for any task, run on GPU if supported
 2
 3 Region * * GPU FBMEM; # for any task, any region, if mapped onto GPU, use FBMEM as default
 4 Region * * CPU SYSMEM; # if mapped onto CPU, use SYSMEM as default
 5
 6 mcpu = Machine(CPU);
 7 mgpu = Machine(GPU);
 8
 9 ========== Above is fixed ==========
10
11 Layout * * * F_order;
```

**Strategy 5**: Align all the regions to 64 bytes while using the Fortran ordering of data

```
 1 Task * GPU,CPU; # for any task, run on GPU if supported
 2
 3 Region * * GPU FBMEM; # for any task, any region, if mapped onto GPU, use FBMEM as default
 4 Region * * CPU SYSMEM; # if mapped onto CPU, use SYSMEM as default
 5
 6 mcpu = Machine(CPU);
 7 mgpu = Machine(GPU);
 8
 9 ========== Above is fixed ==========
10
11 Layout * * * Align==64 F_order;
```

**Strategy 6** Place the task calculate_new_currents onto CPU

```
 1 Task * GPU,CPU; # for any task, run on GPU if supported
 2
 3 Region * * GPU FBMEM; # for any task, any region, if mapped onto GPU, use FBMEM as default
 4 Region * * CPU SYSMEM; # if mapped onto CPU, use SYSMEM as default
 5
 6 mcpu = Machine(CPU);
 7
 8 mgpu = Machine(GPU);
 9
10 Layout * * * SOA C_order;
11
12 ========== Above is fixed ==========
13 Task calculate_new_currents CPU;
```

**Strategy 7**: Collect all the memory used by task calculate_new_currents

```
 1 Task * GPU,CPU; # for any task, run on GPU if supported
 2
 3 Region * * GPU FBMEM; # for any task, any region, if mapped onto GPU, use FBMEM as default
 4 Region * * CPU SYSMEM; # if mapped onto CPU, use SYSMEM as default
 5
 6 mcpu = Machine(CPU);
 7 mgpu = Machine(GPU);
 8
 9 Layout * * * SOA C_order;
10
11 ========== Above is fixed ==========
12 CollectMemory calculate_new_currents *;
```

**Strategy 8**: Ensure that at most 4 tasks of calculate_new_currents can be run at the same time

```
 1 Task * GPU,CPU; # for any task, run on GPU if supported
 2
 3 Region * * GPU FBMEM; # for any task, any region, if mapped onto GPU, use FBMEM as default
 4 Region * * CPU SYSMEM; # if mapped onto CPU, use SYSMEM as default
 5
 6 mcpu = Machine(CPU);
 7 mgpu = Machine(GPU);
 8
 9 Layout * * * SOA C_order;
10
```

```
11 ========== Above is fixed ==========
12 InstanceLimit calculate_new_currents 4;
```

**Strategy 9**: Map the second region argument of task distribute_charge onto GPU's Zero-Copy memory

```
1 Task * GPU,CPU; # for any task, run on GPU if supported
2
3 Region * * GPU FBMEM; # for any task, any region, if mapped onto GPU, use FBMEM as default
4 Region * * CPU SYSMEM; # if mapped onto CPU, use SYSMEM as default
5
6 mcpu = Machine(CPU);
7 mgpu = Machine(GPU);
8
9 Layout * * * SOA C_order;
10
11 ========== Above is fixed ==========
12 Region distribute_charge 1 GPU ZCMEM;
```

**Strategy 10**: Map the tasks of calculate_new_currents,distribute_charge,update_voltages onto GPUs in a 1D cyclic manner: perform a cyclic distribution over both the node and processor dimensions.

```
1 Task * GPU,CPU; # for any task, run on GPU if supported
2
3 Region * * GPU FBMEM; # for any task, any region, if mapped onto GPU, use FBMEM as default
4 Region * * CPU SYSMEM; # if mapped onto CPU, use SYSMEM as default
5
6 mcpu = Machine(CPU);
7 mgpu = Machine(GPU);
8
9 Layout * * * SOA C_order;
10
11 ========== Above is fixed ==========
12 def cyclic1d(Task task) {
13     ip = task.ipoint;
14     # cyclic over node, cyclic over gpu
15     return mgpu[ip[0] % mgpu.size[0], ip[0] / mgpu.size[0] % mgpu.size[1]];
16 }
17
18 IndexTaskMap calculate_new_currents,distribute_charge,update_voltages cyclic1d;
```

