# OpenReview forum: "MAGE: Leveraging LLMs for Automated Mapper Generation in Parallel Programming"
_ICLR.cc/2025/Conference — Submitted to ICLR 2025_

### Official Review · Reviewer_X64Z · 2024-11-01

**Soundness:** 3
**Presentation:** 3
**Contribution:** 2
**Rating:** 5
**Confidence:** 4

**Summary:**

This paper presents an approach to finding the right mapping strategy to map parallel tasks onto processors and memory. The idea is to describe the mapping strategy and tuning parameters like data layouts and memory placement in a domain-specific language (DSL) and then ask LLM to generate a mapping plan in the DSL. The proposed approach uses reinforcement learning to drive the LLM code generation process to provide feedback for the LLM (GPT-40) to improve the correctness of the generated code. The proposed approach was evaluated on a CPU-GPU system using selected HPC workloads. Experimental results show that the proposed approach can match or exceed the performance given by an expert-crafted mapper.

**Strengths:**

This paper provides an interesting study of the use of LLMs to generate high-level code for parallel mapping.

The paper is, in general, easy to follow.

Some interesting results were presented.

**Weaknesses:**

The overall approach appears unnecessarily complex for the problem to be solved. For example, the proposed method requires using a DSL to describe the optimization space, which is likely to be dependent on the specific program being optimized. This, thus, increases engineering overhead and prevents adoption. Additionally, given that only a small set of parameters (processor type, data placement, and data layout) are found to be crucial, why not simply expose these as tunable parameters for classical parameter tuning algorithms? Well-established frameworks like OpenTuner and CompilerGym are designed for performance parameter tuning and could be used for the same purpose.

Similarly, while it may be interesting to test the LLM's capability to generate high-level DSL code, this does not seem to be the right problem for LLMs. The idea of using LLMs to generate a DSL that is then mapped to C/C++ code seems overly complicated and unnecessary.

The evaluation is weak and lacks important details. For example, only three applications were used, but how were these applications implemented? Are they CUDA applications? What CUDA driver and CPU were used in the evaluation? Additionally, why were well-established GPU benchmark suites like Parboil and Rodinia not considered?

**Questions:**

Does the decision procedure described in Figure 4a depend on the program to be optimized? In other words, how general is the decision procedure?

Why not just expose the parameters to a tuning framework? Can you provide a direct comparison?

Why were only three benchmarks used in the evaluation, and how were they implemented?

Please provide details of your evaluation platforms.

---

> ### Author Response · Authors · 2024-11-19
> **Reponse 1**
>
> Thank you for your thoughtful review. We appreciate it!
>
> > Does the decision procedure described in Figure 4a depend on the program to be optimized? In other words, how general is the decision procedure?
>
> The decision procedure in Figure 4a is designed to be general and can be applied to any application program. It takes application-specific metadata, such as the names of tasks and regions, which are extracted from the application source code. Based on this metadata, it generates a mapper tailored to the specific application. This makes the procedure adaptable to various task-based programming applications while maintaining a generalized framework.
>
> > Why not just expose the parameters to a tuning framework? Can you provide a direct comparison?
>
> **We added an experiment comparing our approach with OpenTuner**, where we let OpenTuner decide whether to sample from a set of keywords/parameters. The results shown in Figures 5 and 6 in the updated paper indicate that OpenTuner performs far worse than our agentic LLM-based solution. OpenTuner cannot leverage existing documentation of the DSL or have any prior knowledge of the problem of mapping, while our LLM framework can read the documentation and leverage textual feedback with error explanations and mapping suggestions. This textual feedback enables the LLM to iteratively refine the mapper, addressing issues like GPU memory errors dynamically. In contrast, OpenTuner cannot interpret or act on such textual feedback and often proposes invalid mappers. The ability to form prior knowledge over the decision space, process runtime errors in text form and adapt mappings accordingly makes our approach significantly better.
>
>
> > Why were only three benchmarks used in the evaluation, and how were they implemented?
>
> Our paper evaluates a total of 9 benchmarks. Section 5.2 focuses on 3 scientific applications, and Section 5.3 includes 6 different well-known parallel matrix multiplication algorithms. The simplest benchmark is Stencil, which forms an optimization space of $2^{38}$ possibilities, let alone the most complex application, Pennant, which involves 34 tasks and 11 regions.
>
> The six matrix multiply algorithms that we benchmark have mappers that are as involved as the applications --- parallel matrix multiply algorithms are not simple – based on our estimates, each parallel matrix multiplication algorithm has approximately $10^9$ possible choices for index mapping.
>
> The three scientific applications are developed by domain experts, and six well-known matrix multiplication algorithms are implemented in prior work by experts in performance engineering. The MM algorithms have mappings as complex as the applications – the expert-written mappings for these algorithms are significantly more elaborate than those for scientific applications, requiring roughly twice as many lines of code as the scientific applications.
>
>
>
> > Are they CUDA applications? What CUDA driver and CPU were used in the evaluation? Please provide details of your evaluation platforms.
>
> All nine benchmarks are CUDA applications. The tasks for scientific applications have implementations (i.e., variants) in CUDA for GPU, OpenMP, and CPU to enable the search for processor selection. The matrix-multiplication implementations target GPUs, where the search focuses on different ways to partition and distribute the tensors over multiple GPUs. We conducted experiments using CUDA driver version 12.2 on a platform with two Intel 10-core E5-2640 v4 CPUs, 256 GB of main memory, and four NVIDIA Tesla P100 GPUs.
>
> > Additionally, why were well-established GPU benchmark suites like Parboil and Rodinia not considered?
>
> We chose to focus on the Legion parallel programming framework because it exposes a rich set of correctness-agnostic mapping decisions, enabling performance tuning without affecting application correctness. Other well-established benchmarks do not inherently expose similar abstractions for mapping. Our goal is to demonstrate the feasibility and potential of LLMs for performance tuning in task-based programming systems. We hope this work inspires researchers to extend these ideas to other benchmarks and programming systems in the future.
>
> We hope we have addressed your concerns and questions thoroughly, and we are happy to answer more questions if they arise. Thank you very much!

---

> ### Author Response · Authors · 2024-11-22
> **Hi**
>
> As the author/reviewer discussion period getting close to an end (in 4 days -- Nov 26), we are wondering if 1) Our rebuttal addresses some of your concerns about the paper; 2) there is anything else we can do in the next 4 days to change your opinion on the current rating?

---

> ### Comment · Reviewer_X64Z · 2024-11-25
>
> Thank you for your time in drafting the author's response. Can you automatically generate the DSL program from the target program? This was not clear from the paper, and it appears that this requires user intervention. If this is an automated process that can be applicable to general cases, why not test your approach on well-established GPU benchmarks?
>
> I am not convinced by the argument that parallel matrix multiply algorithms are complex - having a large number of tuning parameter combinations does not necessarily increase the complexity of the program. For example, the number of possible loop-unroll factors for a simple matrix multiplication program can also be large.
>
> I also don't think the comparison to OpenTuner is fair. The number of search iterations given to OpenTuner is too small. As an evolutionary-based algorithm, OpenTuner will require at least thousands of search iterations. Furthermore, I am not sure applying OpenTuner to the DSL is a good idea. Why not just expose the search parameters to OpenTuner, apply the OpenTuner's chosen parameters to the target program, measure the resulting performance and give the measurement as feedback to OpenTuner? While OpenTuner requires a larger number of search iterations, it does not rely on crafting a DSL program and does not require access to the computational resources to run the LLMs.
>
> Given the concerns discussed above, my main reservations still stand and I will keep my original score. However, I do appreciate the author's effort in responding to my feedback and adding the new experimental results. Overall, I feel this is not the right problem for LLMs.

---

> > ### Author Response · Authors · 2024-11-26
> > **Reponse 2**
> >
> > Thanks for your thoughtful reply! We appreciate it.
> >
> > > Can you automatically generate the DSL program from the target program?
> > > Why not test your approach on well-established GPU benchmarks?
> >
> > Sorry if it’s unclear from our prior response. We can automatically generate the DSL program from any target program written in Legion, a task-based programming system [1]. The well-established GPU benchmarks are not implemented on top of the Legion programming abstraction, thus does not expose the correctness-orthogonal mapping decisions. Rewriting the GPU benchmarks in Legion will require wrapping every CUDA kernel into a task in the Legion framework.
> >
> > [1] Bauer et. al. Legion: Expressing Locality and Independence with Logical Regions. SC’12.
> >
> >
> > > I am not convinced by the argument that parallel matrix multiply algorithms are complex
> >
> > Parallel matrix multiplication algorithms are a well-studied problem in the HPC community, with sophisticated algorithms developed by very smart people and implementations heavily optimized by performance experts. Please take a look at these papers [2,3,4,5,6,7] and the implementation [8] before judging whether these algorithms are complex or not. We briefly summarize these algorithms below for your reference.
> >
> > **2D Algorithms** Cannon’s [2] introduced a systolic communication pattern with
> > tiled data partitioning for distributed matrix multiplication. PUMMA [3] and
> > SUMMA [4] extended this approach by supporting non-square matrices and improving communication efficiency through pipelining. They are called 2D algorithms because they partition the matrices into 2D tiles and then map them onto the processor space.
> >
> > **Non-2D Algorithms** Johnson’s [5] introduced a 3D algorithm that partitions the input matrices into 3D tiles and uses additional memory per processor to reduce communication compared to 2D algorithms. Solomonik’s [6] balances between 2D and 3D approaches by using extra memory to further minimize communication. COSMA [7] takes a different approach by optimizing the processor grid and parallelization strategy based on the input size and the machine size.
> >
> > [2] Cannon et. al. A cellular computer to implement the Kalman filter algorithm. 1969.
> >
> > [3] Choi et. al. Pumma: Parallel universal matrix multiplication algorithms on distributed memory concurrent computers. 1994.
> >
> > [4] Van De Geijn et. al. Summa: Scalable universal matrix multiplication algorithm. 1997.
> >
> > [5] Agarwal et. al. A three-dimensional approach to parallel matrix multiplication. 1995.
> >
> > [6] Solomonik et. al. Communication-optimal parallel 2.5d matrix multiplication and lu factorization algorithms. 2011.
> >
> > [7] Kwasniewski et. al. Red-blue pebbling revisited: near optimal parallel matrix-matrix multiplication. 2019.
> >
> > [8] Yadav et. al. DISTAL: The Distributed Tensor Algebra Compiler. 2022.
> >
> >
> > > As an evolutionary-based algorithm, OpenTuner will require at least thousands of search iterations
> >
> > Yes, thanks for providing your intuition and agreeing with us. A traditional evolutionary-based algorithm requires many more iterations to find the better mapper, while our agent-based solution can find better mappers than human experts within 10 iterations. We believe this is an important advancement. Using LLM as optimizers (i.e., “generative optimization”) is a new field of study. Many discrete optimization problems have the potential to be solved by generative optimization (e.g., see applications in mixed integer programming [1]). Our paper demonstrates the powerfulness of generative optimization in the system optimization realm and encourages future work in this area.
> >
> > [1] Ali AhmadiTeshnizi, Wenzhi Gao, and Madeleine Udell. Optimus: Optimization modeling using
> > mip solvers and large language models. ICML, 2024.

---

> > ### Author Response · Authors · 2024-11-26
> > **Reponse 3**
> >
> > > Furthermore, I am not sure applying OpenTuner to the DSL is a good idea.
> >
> > Sorry if there is any misunderstanding in how we conduct the experiments. We do **not** let OpenTuner generate the DSL. Instead, we only expose the enumerable parameters to the OpenTuner. Our search space is as follows:
> >
> >
> > | Name of the Parameter   | Options                               |
> > |--------------------------|---------------------------------------|
> > | Processor Mapping        | CPU, GPU, OpenMP                    |
> > | Memory Selection         | FrameBuffer, System, Zero-Copy      |
> > | Layout Order             | F_order, C_order                    |
> > | Layout Alignment         | None, 64, 128                       |
> > | Index Launch             | None, block1d, cyclic1d, cyclic2d, linearize2d |
> >
> >
> > We implement the search with the EnumParameter class [2,3]. For index launch, OpenTuner need to choose from a set of pre-defined functions because it cannot generate code. There is no code generation in OpenTuner’s search – it’s simply parameter search. Please let me know if you believe there is anything wrong with how we use OpenTuner.
> >
> > [2] https://github.com/jansel/opentuner/blob/master/docs/source/params.rst#enum-parameter
> >
> > [3] https://github.com/jansel/opentuner/blob/master/opentuner/search/manipulator.py#L1026
> >
> >
> > We are happy to answer more questions if they arise. Thank you very much!

---

> > ### Author Response · Authors · 2024-12-02
> > **Hi**
> >
> > With the discussion period coming to a close, do you feel our rebuttal has sufficiently addressed your concerns, or is there anything else you would like us to clarify? Thanks!

---

### Official Review · Reviewer_n6zM · 2024-11-02

**Soundness:** 2
**Presentation:** 2
**Contribution:** 1
**Rating:** 3
**Confidence:** 5

**Summary:**

This paper introduces MAGE, a reinforcement learning-based generator for task mappers, designed to allocate tasks to various devices in distributed environments. The key idea is to propose a domain-specific language (DSL) to represent the mapping strategy, using large language models (LLMs) to produce mapping code in this DSL. Experimental results show that this approach achieves a higher success rate for generating correct mappers and better performance across different benchmarks.

**Strengths:**

1. Proposed a concise DSL representing the task mapping strategies, enabling LLMs to generate valid programs.
2. Used LLM as a heuristic to propose new mapping strategies in a reinforcement learning pipeline.

**Weaknesses:**

1. The work does not introduce new techniques; incorporating an LLM into an existing reinforcement learning framework does not appear to be a novel contribution.
2. There is insufficient insight or evidence explaining why the proposed DSL is well-suited for task mapping representation or LLM generation. The necessity of specific statements and the grammar structure in Figure 3(a) are not clearly explained.
3. Evaluations are limited to simple benchmarks, lacking experiments on realistic applications like device placement for deep learning model training.
4. The correctness of task mappings is not discussed. Some mappings may require inter-device communication to ensure final results' accuracy.

**Questions:**

1. Why is the proposed DSL well-suited for representing task mapping? What is the rationale for the statements in Section 3, and why is the grammar structured as shown in Figure 3(a)?
2. Why does the proposed DSL improve LLM generation? Are there fundamental limitations when using LLMs to generate C++ code? Given that LLMs are typically trained on widely-used languages like C++, might they perform better with more C++ examples in the prompt or constraints to prevent invalid variable names? Alternatively, using a template with fill-in-the-blank prompts in C++ could improve success rates. Thus, the experiments in Section 5.1 don’t entirely prove the DSL’s effectiveness or necessity.
3. Can the two errors shown in Table 1 for your DSL be avoided by improving the prompt?
4. "Mappers do not change the correctness of an application's output; they only affect its performance." This statement may be inaccurate—some mapping strategies require device communication to ensure correctness (e.g., tensor parallelism in Transformer training). How do you verify that the LLM-generated mapper is correct? Are there any formal guarantees? Are communication operations (e.g., all_reduce) handled afterward?
5. One advantage of DSLs is abstracting away low-level parallelization details to generalize across different backends. Is your DSL general enough to apply parallel strategies to other frameworks, like TaskFlow [A]? How much effort is required to support frameworks other than Legion? Would your DSL need adjustments?
6. "Experiments are conducted on a GPU cluster ..." How many GPU nodes were used?
7. Does the underlying hardware affect mapping strategies? For example, if A100 GPUs were used instead of P100s, would the RL pipeline need to be rerun?
8. Can your approach apply to distributed LLM training, and how do the generated task mapping strategies compare with Alpa [B]?
9. Some early RL-based task mappers [C] are not discussed. Could you explain why including LLMs produces better results? Could your method handle the experiments presented in [C]?

[A] Tsung-Wei Huang, Dian-Lun Lin, Chun-Xun Lin, Yibo Lin, "Taskflow: A Lightweight Parallel and Heterogeneous Task Graph Computing System", IEEE Transactions on Parallel and Distributed Systems (TPDS), 2021.

[B] Lianmin Zheng, Zhuohan Li, Hao Zhang, Yonghao Zhuang, Zhifeng Chen, Yanping Huang, Yida Wang, Yuanzhong Xu, Danyang Zhuo, Eric P. Xing, Joseph E. Gonzalez, Ion Stoica, "Alpa: Automating Inter- and Intra-Operator Parallelism for Distributed Deep Learning", OSDI, 2022.

[C] Azalia Mirhoseini, Hieu Pham, Quoc Le, Mohammad Norouzi, Samy Bengio, Benoit Steiner, Yuefeng Zhou, Naveen Kumar, Rasmus Larsen, Jeff Dean, "Device Placement Optimization with Reinforcement Learning", ICML, 2017.

---

> ### Author Response · Authors · 2024-11-19
> **Reponse 1**
>
> Thank you for your thoughtful review. We appreciate it!
>
> > The work does not introduce new techniques; incorporating an LLM into an existing reinforcement learning framework does not appear to be a novel contribution.
>
> Thank you for engaging in a discussion with us. We respectfully disagree. Using LLM as optimizers (i.e., “generative optimization”) is a new field of study. Many discrete optimization problems have the potential to be solved by generative optimization (see applications in mixed integer programming [1]). This is the first paper to show that by using such a technique, we can learn implementations of mappers that surpass previous expert implementations – which has never been shown before. Our paper demonstrates the powerfulness of generative optimization and encourages future work in this area.
>
> [1] Ali AhmadiTeshnizi, Wenzhi Gao, and Madeleine Udell. Optimus: Optimization modeling using
> mip solvers and large language models. arXiv preprint arXiv:2310.06116, 2023.
>
> > Evaluations are limited to simple benchmarks
>
> Please kindly note that the benchmarks are not simple. We have utilized 9 real-world benchmarks in total. Section 5.2 focuses on 3 scientific applications, while Section 5.3 examines 6 well-known parallel matrix multiplication algorithms.
> Even the simplest scientific application, Stencil, is non-trivial: it comprises two tasks and twelve data arguments. Each task and data argument offers two placement options, and each data argument has four additional layout choices, leading to an optimization space of $2^{38}$ possibilities, let alone the most complex application, Pennant, which involves 34 tasks and 11 regions.
> The six matrix multiplication algorithms we benchmark are also intricate. Parallel matrix multiplication algorithms are inherently complex; our estimates indicate that each algorithm has approximately $10^9$ possible choices for index mapping. Moreover, the expert-written mappings for these algorithms are significantly more elaborate than those for scientific applications, requiring roughly twice as many lines of code as the scientific applications.
>
> > The correctness of task mappings is not discussed.
>
> > "Mappers do not change the correctness of an application's output; they only affect its performance." How do you verify that the LLM-generated mapper is correct? Are there any formal guarantees? Are communication operations (e.g., all_reduce) handled afterward?
>
> The mapping is correct by construction: the mapper expressed in the DSL simply cannot give a mapping directive that results in a wrong execution result, because the underlying runtime ensures correct execution.  For example, if two task calls t1(A) and then t2(A) are mapped to different processors of the machine, the runtime will automatically insert the copy of A from where t1 executed to where t2 will execute. All collective operations will be properly handled by the runtime.
>
>
> > Why is the grammar of the DSL structured as shown in Figure 3(a)?
>
> The DSL can succinctly describe the search space (e.g., whether the decision is a per-task, per-region, or per-processor decision) in a syntactically reasonable way, enabling LLMs to generate interpretable programs rather than using discrete numerical numbers to represent each choice. This design simplifies reasoning and ensures code clarity. We are happy to add more explanations of the ideas behind the DSL design in the revised version.

---

> ### Author Response · Authors · 2024-11-19
> **Reponse 2**
>
> > Using a template with fill-in-the-blank prompts in C++; DSL’s effectiveness or necessity
>
> Our goal is not to prove that LLMs cannot generate low-level C++ code correctly, but to highlight that the DSL significantly simplifies the task so that we can let LLMs further explore the search space beyond just generating code given a specified strategy.
>
> **We added a new experiment using fill-in-the-blank prompts in C++.** This approach achieved a 20% success rate (passing the tests for the two memory placement problems out of the ten mapping problems), better than earlier C++ baselines but still far behind the DSL-based generation.
>
> While giving more C++ examples can help LLM with the generation, there is a token limit and context window constraint. In general, code generation for real-world software is known to be a challenging and unsolved problem for LLMs [2,3].
>
> [2] Jimenez et. al. SWE-Bench: Can Language Models Resolve Real-World Github Issues?
>
> [3] Du et. al. Classeval: A Manually-Crafted Benchmark for Evaluating LLMs on Class-Level Code Generation.
>
> To illustrate the effectiveness of DSL, we added Table A1 (lines of code comparison between DSL and C++) to the paper’s appendix. As shown below, for the 9 benchmarks used in our evaluation, the average C++ mapper is 406 lines long, while the average DSL mapper is just 29 lines, a 14x reduction. The DSL abstracts low-level details, making it much easier for LLMs to generate and reason about compared to the complexity of writing hundreds of C++ lines with intricate APIs.
>
> | Application     |   1|   2|   3|   4|   5|   6|   7|   8|   9| Avg.|
> |-----------------|----|----|----|----|----|----|----|----|----|-----|
> | LoC in C++      | 347| 306| 379| 447| 437| 430| 428| 433| 448|  406|
> | LoC in DSL      |  16|  14|  16|  38|  38|  38|  33|  38|  32|   29|
> | LoC Reduction   | 22×| 22×| 24×| 12×| 12×| 11×| 13×| 11×| 14×|  14×|
>
>
> We also updated Figure 3 to better illustrate the code complexity between DSL and C++.
>
> > Can the two errors shown in Table 1 for your DSL be avoided by improving the prompt?
>
> Yes, both errors were syntax-related. We improved our prompt by explicitly emphasizing DSL syntax rules and common pitfalls. With these enhancements, the LLM successfully generates correct DSL mappers on the first trial.
>
> > Is your DSL general enough to apply parallel strategies to other frameworks, like TaskFlow [A]? How much effort is required to support frameworks other than Legion? Would your DSL need adjustments?
>
> Many parallel programming frameworks, including TaskFlow, StarPU, Chapel, X10, and Charm++, designed similar mapping controls. Our DSL covers features like data layout selection, which are not universally supported (e.g., TaskFlow lacks this feature). Adapting the DSL for other frameworks would depend on the number of correctness-agnostic choices exposed by each framework. Adjustments would primarily involve aligning the DSL with the specific controls and abstractions of the target framework.  But we believe the final result would be similar — Legion has the most elaborate mapping API, and other systems would have fewer choices to search.
>
> > How many GPU nodes were used?
>
> We ran all experiments on one node with 4 GPUs.

---

> ### Author Response · Authors · 2024-11-19
> **Response 3**
>
> > Does the underlying hardware affect mapping strategies? For example, if A100 GPUs were used instead of P100s, would the RL pipeline need to be rerun?
>
> Yes, mapping strategies are machine-dependent. Changing the hardware would require rerunning the agentic pipeline to adapt to the new environment. This context-specific nature of mapping underscores the importance of having a fully automated solution to avoid labor-intensive manual tuning.
>
> > Can your approach apply to distributed LLM training, and how do the generated task mapping strategies compare with Alpa?
>
> The "index mapping" search in our method is analogous to the search performed in Alpa for data parallelism and operator parallelism, but our framework does not currently cover pipeline parallelism, which is one aspect of the search space in Alpa for deep learning.
>
> Conversely, Alpa focuses on distributed deep learning workloads, where all computations are typically mapped to GPUs, and thus does not address processor kind selection or memory placement—both of which are essential in our search space for scientific applications. These additional dimensions, such as CPU vs. GPU selection and memory placement and layout optimization, are critical for workloads beyond deep learning.
>
> Our current benchmarks focus on scientific applications and matrix multiplication algorithms. While we do not yet have benchmarks specifically for distributed LLM training, this is an exciting direction for future exploration.
>
> We have added citations [A][C] to the paper (they are highlighted in red). Alpa [B] has already been cited and discussed in Section 3 (we applied red highlight to it as well).
>
> > Some early RL-based task mappers [C] are not discussed. Could you explain why including LLMs produces better results? Could your method handle the experiments presented in [C]?
>
> Thank you for the suggestions! We cited it in the revised version. The intuition behind the advantage of LLM-based methods lies in their ability to read through existing documents on the problem domain to form a better prior over search space and can incorporate textual feedback such as error explanations and suggestions. Traditional RL techniques are much less sample efficient, often resulting in 100x or more search budgets. Our mapping search is general and could be applied to any computation graphs, including those similar to those ML workloads in [C], as part of future work.
>
> **We added an experiment comparing our approach with OpenTuner (traditional RL-based method)**, where we let OpenTuner decide whether to sample from a set of keywords/parameters. The results shown in Figures 5 and 6 in the updated paper indicate that OpenTuner performs far worse than our agentic LLM-based solution. OpenTuner cannot leverage existing documentation of the DSL or have any prior knowledge of the problem of mapping, while our LLM framework can read the documentation and leverage textual feedback with error explanations and mapping suggestions. This textual feedback enables the LLM to iteratively refine the mapper, addressing issues like GPU memory errors dynamically. In contrast, OpenTuner cannot interpret or act on such textual feedback and often proposes invalid mappers. The ability to form prior knowledge over the decision space, process runtime errors in text form and adapt mappings accordingly makes our approach significantly better.
>
>
> We hope we have addressed your concerns and questions thoroughly, and we are happy to answer more questions if they arise. Thank you very much!

---

> ### Author Response · Authors · 2024-11-22
> **Hi**
>
> As the author/reviewer discussion period getting close to an end (in 4 days -- Nov 26), we are wondering if 1) Our rebuttal addresses some of your concerns about the paper; 2) there is anything else we can do in the next 4 days to change your opinion on the current rating?

---

> > ### Comment · Reviewer_n6zM · 2024-11-23
> >
> > Thanks for the detailed response. I appreciate the authors' efforts to improve the paper. However, there are still several issues that remain:
> > 1. The referenced paper [1] appears to be unpublished. While I agree that discrete optimization problems have potential to benefit from generative optimization, it is critical to ensure that the constraints inherent in such problems are carefully encoded into the model. Without this, merely treating an existing LLM as a black-box optimization tool does not significantly advance the field. Heuristic methods can also generate solutions while offering better interpretability. It is important to clarify the rationale for leveraging an LLM in this context.
> > 2. I agree those kernels are not simple in terms of the size of the search space. However, those kernels have been extensively studied in the HPC community, which has already developed high-performance, explainable solutions. As I mentioned, if you could target multi-kernel design (e.g., Transformer models), that would be a more interesting question and more related to the ML audience.
> > 3. "For example, if two task calls t1(A) and then t2(A) are mapped to different processors of the machine, the runtime will automatically insert the copy of A from where t1 executed to where t2 will execute." This approach can be highly inefficient, as not all applications require copying the entire A across devices for execution. Again, a very good example is tensor parallelism for Transformer models. If your framework cannot handle such collective operations and relies solely on the runtime to manage communication, it will require tight coupling with a specific distributed execution framework that supports automatic communication. However, most distributed frameworks (e.g., PyTorch Distributed) rely on users to explicitly insert collective operations. Moreover, leaving correctness guarantees entirely to the runtime can lead to hard-to-debug errors in distributed environments. Correctness guarantees should ideally be ensured at the source level, particularly when the LLM generates DSL code.
> > 4. The comparison of lines of code (LoC) between C++ and the proposed DSL is unconvincing. LLMs can also be used to generate the variables or values to populate placeholders in a C++ template program. In such cases, the effective LoC could be even shorter than that of a DSL. This raises questions about the necessity of introducing a DSL. Following this line, the use of an LLM itself is also questionable, as traditional program synthesis techniques could achieve similar results by filling in C++ templates, provided a proper cost function is defined.
> >
> > Based on concerns regarding correctness, scalability, and motivation, I will maintain my current score.

---

> > > ### Author Response · Authors · 2024-11-26
> > > **Response 4**
> > >
> > > Thanks for your thoughtful review! We appreciate it.
> > >
> > > > The referenced paper [1] appears to be unpublished
> > >
> > > Sorry for not citing it in its published format. The paper “OptiMUS: Scalable Optimization Modeling with (MI)LP Solvers and Large Language Models” is published in  ICML’24: https://proceedings.mlr.press/v235/ahmaditeshnizi24a.html
> > >
> > >
> > > > However, those kernels have been extensively studied in the HPC community, which has already developed high-performance, explainable solutions
> > >
> > > Yes, we think that our result of 31% performance improvement over state-of-the-art is even more impressive given the fact that those matrix multiplication algorithms are such a well-studied problem. AlphaTensor [4] is the first RL system for discovering novel, efficient, and provably correct algorithms for matrix multiplication. Our framework can not only accelerate parallel matrix multiplication algorithms, but also real-world scientific computing applications.
> > >
> > > [4] Fawzi et. al. Discovering faster matrix multiplication algorithms with reinforcement learning. Nature 2022.
> > >
> > > > if you could target multi-kernel design (e.g., Transformer models), that would be a more interesting question and more related to the ML audience.
> > >
> > > Our system does target multi-kernel design. Every benchmark in our evaluation has multiple CUDA kernels, the biggest one has 40+ kernels. Each kernel is a task that can be launched on GPU in Legion, the task-based parallel programming system used in our paper. We argue that Transformer models are a well-optimized workload in the MLSys community, and it can be challenging to further improve their performance. There are existing works published using the Legion [5] programming system (and FlexFlow [6], built on top of Legion) for the DNN training and LLM serving workload [7,8]. Our work does not target those applications.
> > >
> > >
> > > [5] Bauer et. al. Legion: Expressing Locality and Independence with Logical Regions. SC’12.
> > >
> > > [6] Jia et. al. Beyond Data and Model Parallelism for Deep Neural Networks. MLSys’19.
> > >
> > > [7] Unger et. al. Unity: Accelerating DNN Training Through Joint Optimization of Algebraic Transformations and Parallelization. OSDI’22.
> > >
> > > [8] Miao et. al. SpecInfer: Accelerating Large Language Model Serving with Tree-based Speculative Inference and Verification. ASPLOS’24.
> > >
> > >
> > > > If your framework cannot handle such collective operations and relies solely on the runtime to manage communication, it will require tight coupling with a specific distributed execution framework that supports automatic communication.
> > >
> > > Legion can handle collective operations (e.g., all_reduce, all_gather, broadcast, etc.) in an efficient and automatic way. The underlying communication layer is Realm [9]. We do not discuss the implementations of these system details in the paper, because we think the ML audience will not find this relevant or interesting, and this is also not the core contribution of our work. We are happy to include them in the appendix of the final version.
> > >
> > > [9] Treichler et. al. Realm: An Event-Based Low-Level Runtime for Distributed Memory Architectures. PACT’24.
> > >
> > >
> > > > This raises questions about the necessity of introducing a DSL. Following this line, the use of an LLM itself is also questionable, as traditional program synthesis techniques could achieve similar results by filling in C++ templates, provided a proper cost function is defined.
> > >
> > >
> > > We don’t insinuate that DSL is the only way to solve the mapper generation problem. We agree that there are potentially other ways to solve the mapper generation problem. Instead, we want to claim that DSL is an elegant solution. We believe that using LLM agents as optimizers to solve system research problems is an interesting direction, and one challenge is how to define a good interface to leverage LLMs. As we know, application programming interfaces (APIs) in complex system software can be complicated and intended for library developers/expert users. To automate challenges in software systems, language models need **customized agent-system interface**. Our DSL provides such an elegant interface for language models: it’s simple and easy to understand, compact in terms of lines of code.
> > >
> > > Even though other approach (C++ template plus a cost function) might also work as well, we believe our work is pioneering and novel in the optimization problem of system design and suggests its potential for addressing other complex system challenges with LLMs.
> > >
> > >
> > >
> > > We are happy to answer more questions if they arise. Thank you very much!

---

> > > ### Author Response · Authors · 2024-12-02
> > > **Hi**
> > >
> > > As the discussion period winds down, we’d greatly appreciate your feedback—has our rebuttal addressed your concerns, or is there anything further you’d like us to elaborate on?

---

### Official Review · Reviewer_gEAL · 2024-11-03

**Soundness:** 3
**Presentation:** 3
**Contribution:** 3
**Rating:** 6
**Confidence:** 3

**Summary:**

The authors proposed a new DSL to describe/map parallel programming (e.g., to various processors and physical memories) and demonstrated the high-level interface/model makes it efficient for LLM based agents to optimize the codegen of this DSL that reaches expert crafted mappings.

**Strengths:**

* The introduction of a new DSL for LLM to codegen and optimize is intuitive and efficient. LLMs currently are not good at codegen system level compiled languages like C++ (due to lack of context and hallucination) but good at instruction-following structured (and syntax restricted) problems (e.g, math and function-level coding), A DSL significantly regularize the outputs of LLMs

* The use of agentic workflow to automate the iteration and tuning of performant codegen is efficient in catching up with human expert level performance in a much less time/effort

* The experiments and ablation studies make it qualitatively clear DSL is helpful making the codegen functional and the agentic discrete optimization makes it performant

**Weaknesses:**

* The writing can be a bit more verbose, e.g.,:
  - IIUC, the parallel programming is at task/processors level instead of kernel (e.g., CUDA) level, if so it might be worth pointing out earlier in the context (e.g., when people see matrix multiply or GEMM optimization, it is easy to think of kernel instead of distributed GEMM at first impression)
  - The use of "RL" tends to lead people to think of gradient-descent based RL algo instead of agentic based RL flow, a couple explanations in the early context would be helpful to setup the overall pictures

* Ablation study of feedback. The baseline already has performance feedback provided to the agent, a more foundamental baseline IMO would be single or few shot prompting before adapting an agentic flow

**Questions:**

* Is it possible for LLM to discover new GEMM algorithms given target machines, input sizes etc instead of given the algorithms, find the index mapping of iteration/device space as shown in section 5.3?

* IIUC, the index mapping is basically "sharding" in distributed ML frameworks (e.g, Tensor Parallellisms), is it possible to generalize the ideas to automate sharding in a distributed system?

---

> ### Author Response · Authors · 2024-11-19
> **Reponse 1**
>
> Thanks for the thoughtful review! We have uploaded a new rebuttal version. We address your concerns point by point below:
>
> > IIUC, the parallel programming is at task/processors level instead of kernel…
> > The use of "RL" tends to lead people to think of gradient-descent based RL algo instead of agentic based RL flow…
> We have changed our writing in the paper. Please see the updated paper draft where we highlighted the changes in red.
> > Ablation study of feedback: add few-shot prompting before adopting an agentic flow
>
> Thanks for suggesting a new experiment! **We conducted the experiments with zero-shot and few-shot prompting (without feedback)**. The performance of these approaches is much worse than our agent-based solution. We have included these results in Figure 7 of our revised version to highlight the importance of our feedback-driven solution.
>
>
> > Is it possible for LLM to discover new GEMM algorithms?
>
> Thank you for the question! Our results demonstrate that LLMs can find better *implementations* of existing GEMM algorithms. While the algorithms themselves are typically published as pseudo-code in papers, implementing them on distributed machines involves many decisions not well-specified in the pseudo-code. Our LLM-based solution improves the expert implementation of the existing GEMM algorithms.
>
> > Index mapping is basically "sharding" in distributed ML frameworks (e.g., Tensor Parallelisms). Is it possible to generalize the ideas to automate sharding in a distributed system?
>
> Correct. Our "index mapping" is analogous to "sharding" in ML frameworks. While this paper focuses on scientific workloads, the idea of using LLMs to decide sharding is indeed an exciting direction to explore further, especially for distributed machine learning workloads.
>
> We hope we have addressed your concerns and questions thoroughly, and we are happy to answer more questions if they arise. Thank you very much!

---

> > ### Comment · Reviewer_gEAL · 2024-11-20
> >
> > Thanks for addressing my questions. I would like to raise my score if the authors could have demonstrated the same methodology also apply to auto-sharding in distribured ML workloads. However given I am not an expert in scienfitic workloads and couldn't justify the efficacy of it confidently, I would sill keep my current score. Yet I really appreciate the directions the authors have explored.

---

> > > ### Author Response · Authors · 2024-11-22
> > >
> > > Thank you very much for the response. We really appreciate it!

---

### Official Review · Reviewer_G2S5 · 2024-11-04

**Soundness:** 3
**Presentation:** 3
**Contribution:** 3
**Rating:** 6
**Confidence:** 3

**Summary:**

This paper introduces a new automated mapper generation framework (MAGE) that utilizes large language models (LLMs) and a domain-specific language (DSL) to generate efficient mappers. The research aims to address the high costs and time consumption associated with manually crafting traditional mappers. By introducing reinforcement learning (RL), the paper demonstrates that the generated mappers can significantly improve performance in multiple benchmark tests compared to expert-written mappers, achieving up to 34% acceleration and improve the throughput of parallel matrix multiplication algorithms by up to 31%, reducing development time from several days to just a few minutes.

**Strengths:**

1. The introduction of the DSL simplifies the complexity of mapper generation and improves the efficiency of the generation process.

2. The reinforcement learning feedback mechanism enhances the performance optimization capabilities of the mappers.

3. Experimental results show that the generated mappers outperform expert-written mappers in terms of performance, validating the effectiveness of the proposed method.

**Weaknesses:**

1. Limited Experimental Scope:

   In Section 5, the authors primarily focus on benchmarks such as "circuit simulation," "stencil computation," and "Pennant." The selection of these applications is relatively narrow and lacks testing across other types of parallel computing tasks. Failing to cover a broader range of application scenarios may limit the generalizability and reliability of the proposed method.

2. Insufficient Discussion on Mapping Strategy Selection

   Although the authors propose the generation of multiple mapping strategies, Section 4.2 lacks a thorough analysis of the specific impacts of different strategy choices. For example, when discussing decisions such as "task selection" or "memory allocation," there is no exploration of how these decisions may affect final performance in various scenarios.

3. Lack of Discussion on Failure Cases

   In Section 5, while the authors present successful cases, there is minimal discussion regarding instances where the generated mappers failed in certain tests. The authors do not provide an analysis of the reasons for these failures, which is a significant oversight. Understanding the failure modes could offer valuable insights into the limitations of the proposed method and help identify areas for improvement. This lack of analysis may lead to an incomplete understanding of the robustness and reliability of the mapper generation approach in practical applications.

**Questions:**

1. In Section 4, the authors introduce the DSL designed for mapper generation. The text states that "By providing a higher-level abstraction than C++ APIs, the DSL simplifies interfacing with LLMs," but it does not elaborate on how this abstraction specifically addresses the complexities of C++ APIs and low-level details. Additionally, there is no discussion about whether using this DSL results in the loss of low-level details inherent to C++, which could be crucial for performance optimization and effective resource management. With such a high level of abstraction, will using the DSL result in the loss of low-level C++ details?

2. In Figure 7(c), the Perf+Err line and the Full Feedback Mode line exhibit alternating performance increases over 10 iterations. Is there a need for more iterations to ensure that the Full Feedback Mode consistently outperforms the others?

3. In Figure 7, the error explanation is provided when errors occur due to inappropriate mapping decisions or syntax errors. Why do such explanations have a significant impact on the final throughput results?

---

> ### Author Response · Authors · 2024-11-19
> **Reponse 1**
>
> Thank you for your thoughtful review! We uploaded a revised version of the paper. We address your concerns point by point as follows:
>
> > Limited Experimental Scope, only 3 applications
>
> Our paper evaluates a total of 9 benchmarks. Section 5.2 focuses on 3 real-world scientific applications, while Section 5.3 includes 6 different well-known parallel matrix multiplication algorithms. The six matrix multiply algorithms that we benchmark have mappers that are as involved as the applications --- parallel matrix multiply algorithms are not simple. Based on our estimates, each parallel matrix multiplication algorithm has approximately 10^9 possible choices for index mapping.
>
> > Insufficient Discussion on Mapping Strategy Selection in Section 4.2.
>
> > Decisions such as "task selection" or "memory allocation," there is no exploration of how these decisions may affect final performance
>
>
> The performance impact of different mapping strategies is discussed in Section 3 and Section 4.1, along with the design of our DSL.
>
> For task selection, we are selecting which processor (GPU or CPU) to process the task. This is determined by factors such as GPU memory and kernel execution time. It is a per-task decision. This choice depends on factors such as task size, GPU memory, and kernel launch latency. For example, tiny tasks that require very little computation may prefer to run on CPUs due to the GPU kernel launch overhead, whereas tasks with large memory footprints might prefer to be assigned to OpenMP or CPU when GPU memory is insufficient.
>
>
> For memory placement, data can be placed in GPU’s FrameBuffer for faster access, or to ZeroCopy memory for shared access between CPU and GPU (to avoid data communication cost between CPU and GPU), or in CPU system memory for large data (because GPU memory is usually more limited). Each choice introduces a trade-off between memory access speed, memory usage, and transfer overhead.
>
>
> > Lack of Discussion on Failure Cases
>
> Apologies for not emphasizing this sufficiently. We analyze failure cases in Section 5.1, particularly focusing on compilation issues and test failures. In our revised version, we highlight this discussion more clearly and structure it better. Please see the revised version in detail. Here is the summarized failure case analysis:
>
> Compilation errors in C++ are caused by LLM’s failure to understand contextual dependencies in real-world software such as hallucinating non-existent variables.
> When code compiles but fails the test case, this happens because LLMs fail to understand how to coordinate different APIs together to achieve certain functionality. A detailed example is provided in the paper.
> For DSL, the failure case stems from incorrect syntax.
>
> > How does the DSL specifically address the complexities of C++ APIs and low-level details?
>
> We included compiler implementation details in the appendix (Sections A2, A3, A4) and provided the complete compiler code in the supplementary material.
>
> > With such a high level of abstraction, will using the DSL result in the loss of low-level C++ details?
>
>
> DSL is more restricted than what can be done at the C++ level, but it still can express better mappers than the hand-written C++ designed by experts. In our experiment results, all the mappers that are found to be better than expert-written C++ mappers are expressible in the DSL.
>
> > In Figure 7(c), the second best mode line and the best mode line exhibit alternating performance increases over 10 iterations. Is there a need for more iterations to ensure Full Feedback Mode consistently outperforms others?
>
> Thank you for pointing this out! **We extended experiments to run up to 15 iterations for Figure 7(c)**. See the updated Figure 7(c) in the revised version of the paper. The result showed after 10 iterations, the performance of two modes converged.
>
> We have expected this result. We highlight that the Full Feedback Mode is most useful to provide efficient learning signals early on when errors occur, by suggesting how the LLMs can correct the execution error. It does not provide useful suggestions on how to improve the performance of the mapper. We show this phenomenon in both Figure 7(a) and 7(b), where the Full feedback mode (System + Explain + Suggest) gained higher normalized throughput much faster than other modes.

---

> ### Author Response · Authors · 2024-11-19
> **Reponse 2**
>
> > Why do error explanations significantly impact final throughput results?
>
> Error explanations help LLMs interpret and fix mistakes more effectively in subsequent iterations. Raw error messages alone (without explanation) lead to repeated mistakes because the raw error messages can be hard to interpret (e.g., LLMs cannot understand what this raw error message mean: “Assertion failed: stride does not match expected value.”, but the error explanation “Memory layout is unexpected” is more informative). We added more examples of enhanced feedback in our revised version. We show different kinds of feedback below, with more examples in the appendix (Table A2) of our updated paper.
>
>
> | Mapper         | System Feedback                                                                   |       Enhanced Feedback      |                                                                           |
> |----------------|-----------------------------------------------------------------------------------|:----------------------------:|---------------------------------------------------------------------------|
> |                |                                                                                   | Explain                      | Suggest                                                                   |
> | mapper1 | Compile Error: Syntax error, unexpected :, expecting {                  | N/A                          | There should be no colon : in function definition.               |
> | mapper2 | Execution Error: Assertion failed: stride does not match expected value. | Memory layout is unexpected. | Adjust the layout constraints or move tasks to different processor types. |
> | mapper3 | Performance Metric: Execution time is 0.03s.                             | N/A                          | Move more tasks to GPU to reduce execution time.                          |
>
>
> We hope we have addressed your concerns and questions thoroughly, and we are happy to answer more questions if they arise. Thank you very much!

---

> ### Author Response · Authors · 2024-12-02
> **Hi**
>
> As we approach the end of the discussion period, we were wondering if there are any remaining concerns or questions about our paper that we could address?

---

### Meta-Review · Area_Chair_g2Z3 · 2024-12-11

**Metareview:**

This paper studied techniques for mapper generation based on a combination of RL methods and LLMs.  Overall, there are a number of positive and negative points raised, but the latter slightly outweighing the former and being present in all reviews (even those with a score slightly above the threshold) and persisting after the discussion period.  Eventually, none of the reviewers were willing to champion the paper, and 1-2 reviewers still maintained the rejection decision.  The concerns raised included lack of generalization, limited experiments, issues with clarity and complexity, and more fundamentally, the suitability of the problem for current LLMs.

**Additional Comments On Reviewer Discussion:**

The authors gave detailed responses and most of the reviewers replied to the rebuttal at least once.  Apart from the forum replies, one also confirmed their decision via email.

---

### Decision · Program_Chairs · 2025-01-22

Reject